# Efficacy profile of the CYD-TDV dengue vaccine revealed by Bayesian survival analysis of individual-level phase III data

Daniel J Laydon[1]*, Ilaria Dorigatti[1], Wes R Hinsley[1], Gemma Nedjati-Gilani[1], Laurent Coudeville[2], Neil M Ferguson[1]

[1]MRC Centre for Global Infectious Disease Analysis, School of Public Health, Imperial College London, Faculty of Medicine, London, United Kingdom; [2]Sanofi Pasteur, Lyon, France

## Abstract

**Background:** Sanofi-Pasteur's CYD-TDV is the only licensed dengue vaccine. Two phase three trials showed higher efficacy in seropositive than seronegative recipients. Hospital follow-up revealed increased hospitalisation in 2–5- year-old vaccinees, where serostatus and age effects were unresolved.

**Methods:** We fit a survival model to individual-level data from both trials, including year 1 of hospital follow-up. We determine efficacy by age, serostatus, serotype and severity, and examine efficacy duration and vaccine action mechanism.

**Results:** Our modelling indicates that vaccine-induced immunity is long-lived in seropositive recipients, and therefore that vaccinating seropositives gives higher protection than two natural infections. Long-term increased hospitalisation risk outweighs short-lived immunity in seronegatives. Independently of serostatus, transient immunity increases with age, and is highest against serotype 4. Benefit is higher in seropositives, and risk enhancement is greater in seronegatives, against hospitalised disease than against febrile disease.

**Conclusions:** Our results support vaccinating seropositives only. Rapid diagnostic tests would enable viable 'screen-then-vaccinate' programs. Since CYD-TDV acts as a silent infection, long-term safety of other vaccine candidates must be closely monitored.

**Funding:** Bill & Melinda Gates Foundation, National Institute for Health Research, UK Medical Research Council, Wellcome Trust, Royal Society.

**Clinical trial number:** NCT01373281 and NCT01374516.

*For correspondence:
d.laydon@imperial.ac.uk

## Introduction

Over 40% of the world population is at risk of dengue infection. An estimated 105 million infections and approximately 50 million symptomatic cases occur each year (*Stanaway et al., 2016*; *Cattarino et al., 2020*). Dengue disease is caused by four distinct viruses, termed serotypes (DENV-1–4). Infection confers lifelong immunity to a homologous serotype, but against a heterologous serotype protective immunity is only temporary (*St John and Rathore, 2019*). Furthermore, secondary infection with a heterologous serotype drastically increases the likelihood of disease (*St John and Rathore, 2019*).

Traditional vector control interventions have had little impact on dengue disease burden (*Guzman et al., 2010*) and no antiviral treatments yet exist. Several vaccine candidates are in development, but the only licensed vaccine is Sanofi-Pasteur's CYD-TDV (marketed as *Dengvaxia*). CYD-TDV is a live attenuated tetravalent chimeric vaccine, where genes for the structural proteins (E and prM) are taken from the four DENV serotypes, while the other proteins are based on the yellow fever

17D vaccine strain. The vaccine has now been licensed in 21 countries and the EU. A phase two trial in 2012 (ClinicalTrials.gov number NCT00842530) (*Sabchareon et al., 2012*) reported moderate efficacy of 30.2% (−13.4% to 56.6%) and showed the vaccine to be well tolerated and largely safe. Two large scale phase three trials followed: the CYD14 trial (ClinicalTrials.gov number NCT01373281) in South East Asia of 10,275 children aged 2–14 (*Capeding et al., 2014*), and the CYD15 trial (ClinicalTrials.gov number NCT01374516) in Latin America of 20,869 children aged 9–16 (*Villar et al., 2015*). After stratifying by age, participants were randomly assigned to vaccine or control arms in a 2:1 ratio, and vaccine doses were given at baseline then 6 and 12 months later. For a subset of participants (approximately 20% for CYD14% and 10% for CYD15), immunogenicity and prior dengue exposure was determined using baseline sera. Participants were actively surveilled by weekly phone calls for 25 months post-first dose (where any symptomatic disease was detected), after which surveillance was passive using routine hospital surveillance, (where only hospitalisations were detected). See (*Capeding et al., 2014*; *Villar et al., 2015*; *Hadinegoro et al., 2015*) for further details of the trial design.

The CYD14 and CYD15 trials showed overall vaccine efficacies of 56.5% (43.8%–66.4%) and 60.8% (52.0%–68.0%) respectively, with efficacy varying significantly by serotype and prior exposure. However, in 2015, results from the first year of long-term follow up (*Hadinegoro et al., 2015*) showed that while the vaccine remained beneficial overall, the number of hospitalisations among 2–5 year olds was significantly greater in vaccinees than in controls. A potential explanation for these results considered age as a proxy for serostatus (*Aguiar and Stollenwerk, 2018*), and that the vaccine may act as a 'silent' disease-free infection that primes host immunity (*Ferguson et al., 2016*; *Flasche et al., 2016*). Therefore, a seronegative child, who would ordinarily experience their first and relatively low-risk natural infection, would after vaccination instead experience a 'secondary-like' infection that is more predisposed to clinically apparent disease. Conversely, a child with a single prior natural infection would have a lower risk of disease when exposed to dengue post-vaccination, normally associated with tertiary and quaternary infection [*Figure 1*].

The immunogenicity subset was only a small fraction of the entire trial, and the estimated efficacy in seronegatives had wide confidence intervals indicating neither benefit nor harm, and so it was not possible to determine conclusively whether age or lack of prior exposure was the dominant factor in the increased hospitalisation of 2–5 year-old vaccinees. Further, it was not possible to retroactively expand the immunogenicity subset to determine prior dengue exposure, as *Dengvaxia* can elicit

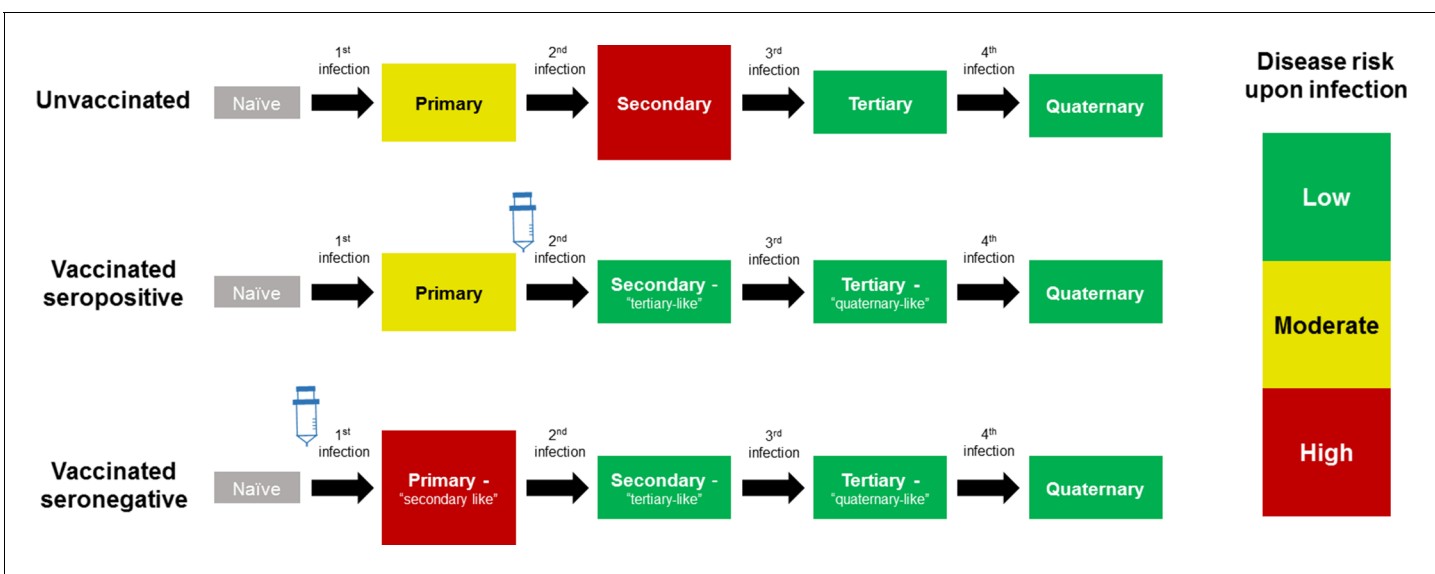

**Figure 1.** Model Schematic 1: vaccine as silent infection. Top row: unvaccinated individuals are naive, or infected with either: their first dengue infection with moderate disease risk; their second higher risk infection; or their third or fourth lower risk infection. Middle row: vaccinating seropositive individuals with a single prior infection lowers the disease risk associated with secondary infection to that associated with tertiary and quaternary infection. Bottom row: vaccinating naive (seronegative) individuals increases disease risk upon their first natural infection. Figure adapted from *Ferguson et al., 2016* with permission.

antibody responses that would test positive under a plaque reduction neutralisation test (PRNT). Therefore, and because the vaccine showed the greatest benefit in children aged nine or older, the vaccine was licensed for use above this age, independent of baseline serostatus.

An ELISA assay detecting anti-dengue non-structural protein 1 (NS1) IgG antibodies then provided a novel approach to retrospectively assess the serostatus of trial participants prior to vaccination (*Nascimento et al., 2018*). *Dengvaxia* expresses yellow fever NS1, not dengue NS1, and therefore this assay can distinguish between natural infection and exposure to the vaccine. Blood samples of trial participants who contracted virologically confirmed dengue during follow-up were analysed using the NS1 assay in the CYD14 and CYD15 trials. The results provided clear evidence of the enhanced risk of hospitalised or severe dengue disease in baseline seronegative vaccinees (*Sridhar et al., 2018*) and these were in line with refined estimates of vaccine efficacy obtained with machine learning (*Dorigatti et al., 2018*). Subsequently, in November 2017, *Dengvaxia* was recommended only in persons with a confirmed prior dengue infection (*WHO, 2018*).

Here we present a survival model with time and age varying hazards, which we fit to the individual level phase three CYD14 and CYD15 data, up to and including the first year of long-term follow-up (*Hadinegoro et al., 2015*). We characterize the efficacy profile and mode of action of the vaccine, which we find to be consistent with the 'vaccine as silent infection' hypothesis. We refine previous estimates, and examine the vaccine's duration of protection and its efficacy against both febrile dengue disease and hospitalized disease. Our results provide a comprehensive characterization of CYD-TDV's safety and efficacy, and demonstrate the need for long-term follow up in the phase three trials of other dengue vaccine candidates currently in development.

## Materials and methods

### Data

We use the individual-level trial data from the CYD14 and CYD15 trials, in both the active phase (25 months post first dose) and the 1$^{st}$ year of passive phase hospital follow-up. In the active phase, all symptomatic dengue disease is detected, but in the passive phase only hospitalisations were detected. All cases refer to virologically confirmed dengue. Infecting serotype is known for almost all cases (97.6% for CYD14, 95.8% for CYD15, 96.7% overall). Baseline serostatus is known for only a minority of subjects (19.3% for CYD14, 9.6% for CYD15, 12.7% overall). Model variants (including our main model) that consider serotype-specific effects omit all cases of unknown serotype. We right-censor after date of first case for each patient, and so do not consider multiple cases per patient.

### Model

We divide trial participants by trial arm *a* and baseline serostatus *b* (0 = seronegative or 1 = seropositive at baseline), as described in *Figures 1* and *2*. Disease risk is allowed to vary by the number of prior dengue exposures and by disease type (where disease type refers either to trial phase (active = 0; passive = 1) or disease severity (non-severe/non-hospitalised = 0, severe/hospitalised = 1)). We consider a country-specific baseline hazard of disease (or force of infection, i.e. the risk of disease among susceptibles) as a spline $\lambda_c(t)$, and we link baseline seropositivity to each participant's age and the background transmission intensity in their country (see below). A trial participant of age $\alpha$ in trial arm *a*, with baseline serostatus *b*, in country *c* is subject to the following hazard from serotype *d* of disease type *D* at time *t*:

$$\lambda_{abcdD}(t,\alpha) = \lambda_c(t)\rho_{cd}M_b Z(\alpha)R_{abcD}(\alpha)\big(1 - \delta_{a,Vac}I^*_{bd}(\alpha,t,t_F)\big)$$

where $R_{abcD}(\alpha)$ is the relative risk of disease associated with natural infection. $M_b$ is the multiplier of the baseline hazard associated with baseline serostatus *b*, equal to 1 for seronegatives and fitted for seropositives. This parameter reflects seropositive participants' reduced infection risk due to their immunity to at least one serotype.

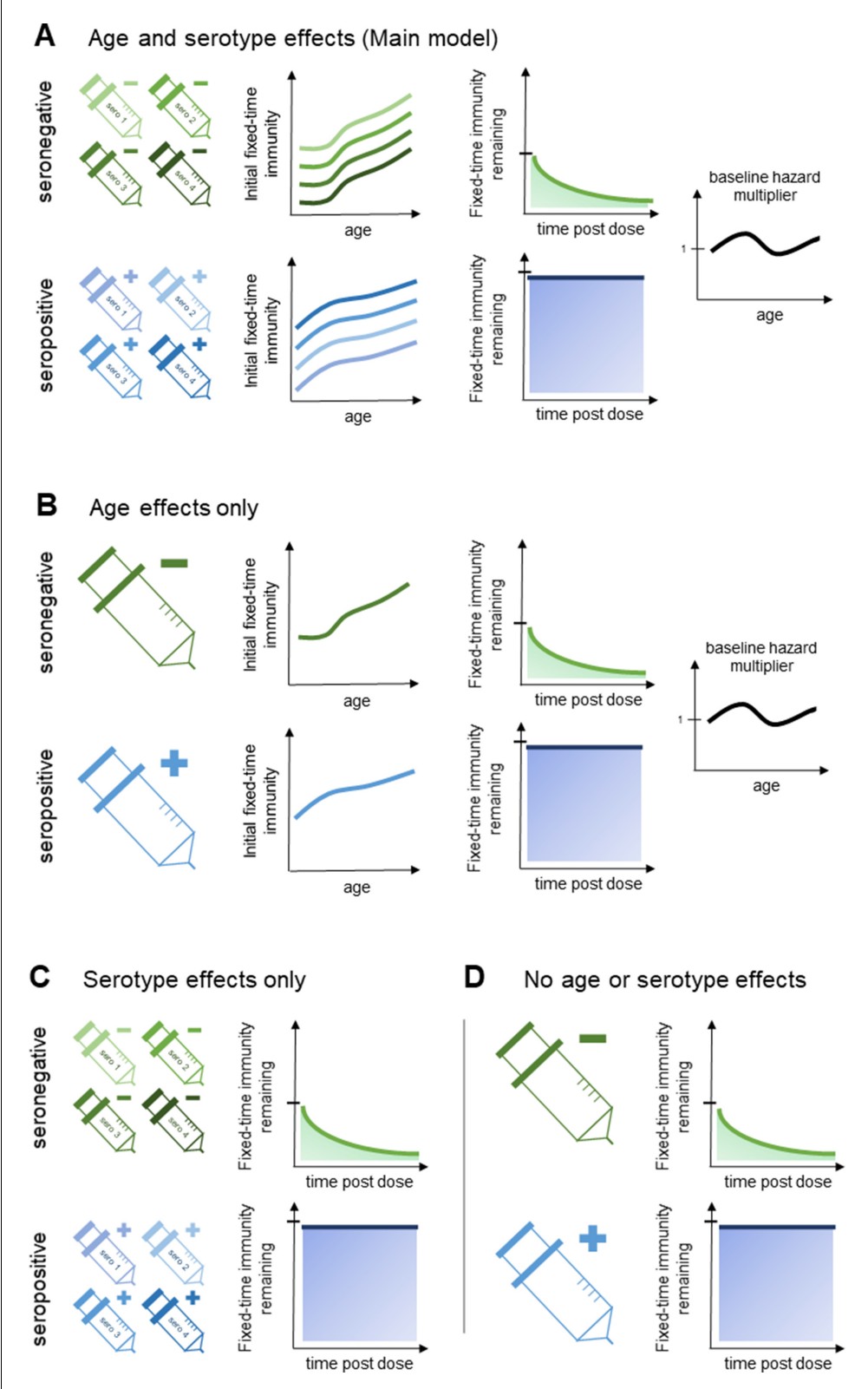

**Figure 2.** Model schematic 2: vaccine-induced transient immunity model hierarchy. In all model variants, we allow the initial magnitude and mean duration of transient immunity to vary by baseline serostatus (exponential waning is assumed). (A) Our main model allows transient immunity magnitude to vary by age and serotype, as well as baseline serostatus. For each serostatus, we model transient immunity with age, and serotype effects are incorporated additively (see Materials and methods for details). We further include an age-specific multiplier of the baseline hazard. (B) and (C) show

*Figure 2 continued*

reduced model variants that dispense with serotype and age effects, respectively. (D) Shows our simplest model variant without explicit age or serotype effects.

$\rho_{cd} \in [0,1]$ is the proportion of serotype $d$ in country $c$ (assumed to be constant, with $\sum_{d=1}^{4} \rho_{cd} = 1$, see below); $Z(\alpha)$ is the multiplier of the force of infection for age $\alpha$; $\delta_{\alpha Vac}$ is the Kronecker delta defined by

$$\delta_{a,Vac} = \begin{cases} 1 & a = \text{Vaccine} \\ 0 & a = \text{Control} \end{cases}$$

We define the transient immunity $I_{bd}{}^*(\alpha, t, t_F)$ against serotype $d$ for vaccinees of age $\alpha$ and baseline serostatus $b$ at time $t$, given time of most recent vaccine dose $t_F$ by

$$I_{bd}^*(\alpha, t, t_F) = \begin{cases} I_{bd}(\alpha) \exp(-(t - t_F)/\tau_b) & I_{bd}(\alpha) > 0 \\ I_{bd}(\alpha) & \text{Otherwise} \end{cases}$$

That is, positive values of transient immunity wane exponentially (to reflect previously observed antibody dynamics *Clapham et al., 2016*). Additional details on age-specific transient immunity $I_{bd}(\alpha)$ and duration $\tau_b$, and force of infection $Z(\alpha)$ are given below [*Figure 2A*].

The relative risk of disease of type $D$ for a subject with $i$ prior dengue exposures is given by $K_{iD}$. $K_{1\,0} = 1$ (secondary febrile dengue illness) is taken to be the baseline, and we assume the relative risk is the same for tertiary and quaternary infection of either type, and so $K_{2D} = K_{3D}$. Our model considers serostatus to be binary (either seropositive or seronegative), and so we define the risk of disease $\varphi_{cD}(\alpha)$ among seropositive participants, which is an aggregate of the risks of disease given monotypic or multitypic infection history, shown below.

Our main model considers vaccination to alter the risk of disease associated with prior exposure, by acting as a silent, disease-free infection *(Figure 1)*. Therefore, the relative risks are defined as follows:

$$R_{abc0}(\alpha) = \begin{cases} K_{0,0} & a = \text{Control}, \quad b = 0 \\ \varphi_{c0}(\alpha) & a = \text{Control}, \quad b = 1 \\ K_{1,0} & a = \text{Vaccine}, \quad b = 0 \\ K_{2,0} & a = \text{Vaccine}, \quad b = 1 \end{cases}$$

when considering the active phase of the trial (or symptomatic disease of any severity) and

$$R_{abc1}(\alpha) = \begin{cases} K_{0,0}K_{0,1} & a = \text{Control}, \quad b = 0 \\ \varphi_{c0}(\alpha)\varphi_{c1}(\alpha) & a = \text{Control}, \quad b = 1 \\ K_{1,0}K_{1,1} & a = \text{Vaccine}, \quad b = 0 \\ K_{2,0}K_{2,1} & a = \text{Vaccine}, \quad b = 1 \end{cases}$$

when considering the passive phase of the trial or hospitalised/severe disease. A glossary of the above terms can be found in *Supplementary file 1*.

## Severity analysis

We interpret relative risks in two different ways depending on our analysis. The probability of case detection depends on the degree of trial surveillance, and so by default we allow relative risks to differ between the active and passive trial phases. For example, $K_{i = 2, D = 0}$ is the risk of clinically apparent disease (of any severity) in the active phase for those with two or more prior infection. Alternatively, when distinguishing between severe and non-severe disease (or equivalently between hospitalised and non-hospitalised disease), risk of disease does not differ between trial phases but between disease severities, for example, the risk of severe disease in seronegatives would be $K_{i = 0, D = 1}$. In practice, these two interpretations of the relative risk parameters are largely equivalent as non-hospitalised disease is detected exclusively by active surveillance and passive surveillance only detects hospitalised or severe disease. It does however change the calculation of survival probabilities. We allow relative risks of hospitalised disease to differ between CYD14 and CYD15 trials to

account for the non-standardised hospitalisation criteria between Southeast Asia and Latin America. We do not do so when modelling severe disease since the trials used the WHO dengue severity criteria to ascribe disease severity in all trial sites. We fix the ratios $K_{0,1}/K_{1,1} = K_{2,1}/K_{1,1} = 0.25$ (*Ferguson et al., 2016*), while the proportion of symptomatic secondary infections that require hospitalisation (or that result in severe disease) $K_{1,1}$ are fitted parameters.

It should be stressed that in our formulation overall vaccine benefit cannot be measured by transient immunity alone, but rather in combination with the change in relative risk induced by vaccination.

## Baseline hazard splines

We model the baseline hazard for each country as a quadratic spline. We divide the follow-up period of length $T$ into $n$-1 intervals $\{t_k\}_{k=1}^n$ with $t_k = k \times \frac{T}{n}$, and define $\lambda_c(t_k) = \kappa_{ck}$ as the knots of the spline for country $c$.

$$\lambda_c(t) = \max\left\{ 0, \begin{cases} \sum_{i=0}^2 \beta_{ick}t^i; & t_k \leq t < t_{k+1}; \quad k=1,2,...,n-3 \\ \sum_{i=0}^2 \beta_{ic(n-2)}t^i; & t_{n-2} \leq t < t_n \\ \kappa_{c1}; & t \leq t_1 \\ \kappa_{cn}; & t \geq t_n \end{cases} \right\}$$

The observation period is approximately $T$ = 4 years, and we use $n$ =10 knots, spaced at 4-month intervals. For polynomial $\sum_{i=0}^2 \beta_{ick}t^i$, we solve the following equations for coefficients $\{\beta_{0ck}, \beta_{1ck}, \beta_{2ck}\}$:

$$\lambda_c(t_k) = \kappa_{c(k)} = \sum_{i=0}^2 \beta_{ick}t_k^i$$

$$\lambda_c(t_{k+1}) = \kappa_{c(k+1)} = \sum_{i=0}^2 \beta_{ick}t_{k+1}^i$$

$$\lambda_c(t_{k+2}) = \kappa_{c(k+2)} = \sum_{i=0}^2 \beta_{ick}t_{k+2}^i$$

The knot locations $\{t_k\}_{k=1}^n$ are fixed and their values $\{\kappa_{ck}\}_{k=1}^n$ are fitted parameters.

## Relative risk in seropositives

If $h_c$ is the constant historical force of infection in country $c$, then the probability of remaining seronegative until age $\alpha$ is given by

$p_{0c}(\alpha) = e^{-h_c \times \alpha}$.

Therefore, the probability of seropositivity (i.e. at least one infection) by age $\alpha$ is given by $1 - e^{-h_c \times \alpha}$. Assuming that each serotype carries an equal force of infection, then the probability of exactly one infection with any serotype is given by

$$p_{1c}(\alpha) = 4 \times e^{-3h_c/4 \times \alpha} \times \left(1 - e^{-h_c \alpha/4}\right)$$

The relative risk of disease $\varphi_{cD}(\alpha)$ in seropositive participants is therefore a weighted average of the risk in participants with one or more than one prior exposure.

$$\varphi_{cD}(\alpha) = \frac{p_{1c}(\alpha)}{1 - p_{0c}(\alpha)}K_{1D} + \left(1 - \frac{p_{1c}(\alpha)}{1 - p_{0c}(\alpha)}\right)K_{2D}$$

Note that the historical hazards $h_c$ refer to infection, not disease, and that this approximation assumes that historical force of infection is equal across serotypes.

## Serotype proportions

The proportions $\rho_{cd}$ of serotype $d$ in country $c$ must satisfy $\sum_{d}^{4}\rho_{cd}=1$ for all $c$. Therefore, given proportions for three serotypes, the fourth is explicitly determined. We fit three parameters $q_{cy}$ ($y = 1, 2, 3$) for each country $c$ and calculate

$$\rho_{c1} = q_{c1}$$

$$\rho_{c2} = q_{c2} \times (1 - q_{c1})$$

$$\rho_{c3} = q_{c3} \times (1 - q_{c2}) \times (1 - q_{c1})$$

$$\rho_{c4} = (1 - q_{c3}) \times (1 - q_{c2}) \times (1 - q_{c1})$$

Each parameter $q_{cy}$ is fitted with prior $Unif(0,1)$.

## Age effects

We use a step function to model age-specific transient immunity and force of infection multiplier. This function is constant within the age groups 2–5, 6–11 and 12–16 years. We also considered a quadratic spline formulation, similar to the baseline hazard, with four knots placed at ages 2, 6, 12 and 16 years, although this did not sufficiently improve model fit.

We model serotype and age effects additively (*Figure 2A*), that is, if $A_b(\alpha)$ gives the relationship between transient immunity and age for serostatus $b$, then the (initial) magnitude of transient immunity $I_{bd}(\alpha)$ for baseline serostatus $b$ and serotype $d$ for age $\alpha$ is given by

$$I_{bd}(\alpha) = A_b(\alpha) + s_{bd}$$

where $s_{bd}$ is the intercept for baseline serostatus $b$ and serotype $d$ (fixed at 0 for serotype $d = 1$).

## Likelihood

If the hazard due to all serotypes combined is given by

$$\lambda_{abcD}(t,\alpha) = \sum_{d=1}^{4} \lambda_{abcdD}(t,\alpha)$$

then where relative risks distinguish between trial phases, we let

$$\lambda^*_{abc}(t,\alpha) = \begin{cases} \lambda_{abc0}(t,\alpha) & t < t_P \\ \lambda_{abc1}(t,\alpha) & t \geq t_P \end{cases}$$

where $t_P$ is the date that active surveillance ends and passive surveillance begins, and define the integrated hazard between start and end times $t_S$ and $t_E$ as

$$\Lambda_{abc}(t_S, t_E, \alpha) = \int_{t_S}^{t_E} \lambda^*_{abc}(t,\alpha) dt$$

Our model does not consider multiple disease episodes for the same patient over the observation period, and subjects are right-censored after they become a case for the first time. Therefore, when relative risks distinguish between the severity of disease, 'survival' between times $t_S$ and $t_E$ refers to surviving disease of both severities, and we therefore define the integrated hazard additively as

$$\Lambda_{abc}(t_S, t_E, \alpha) = \int_{t_S}^{t_E} (\lambda_{abc0}(t,\alpha) + \lambda_{abc1}(t,\alpha)) dt$$

This interpretation assumes that hazards are proportional between disease severities (although not between trial arms, countries or between number of prior infections).

In both formulations, the probability $\mathbb{P}_{abc}(t_S, t_E, \alpha)$ of remaining disease free from between times $t_S$ and $t_E$ is given by

$$\mathbb{P}_{abc}(t_S, t_E, \alpha) = \exp(-\Lambda_{abc}(t_S, t_E, \alpha))$$

and so the probability $\mathbb{Q}_{abcdD}(t_S, t_E, \alpha)$ of disease from serotype $d$ of type $D$ at time $t_E$ is given by

$$\mathbb{Q}_{abcdD}(t_S, t_E, \alpha) = \mathbb{P}_{abc}(t_S, t_E, \alpha)\lambda_{abcdD}(t_E, \alpha) \times T_I$$

where $T_I$ is the time interval within which the hazard is assumed to be constant. We take $T_I$ to be 1 day.

For brevity, we combine the above to denote the probability of clinical outcome $C$ given parameters $\theta$ by

$$\mathbb{P}^*_{abc}(C, t_S, t_E, \alpha; \theta) = \mathbb{P}^*_{abc}(C, t_S, t_E, \alpha) = \begin{cases} \mathbb{Q}_{abcdD}(t_S, t_E, \alpha) & C = case \\ \mathbb{P}_{abc}(t_S, t_E, \alpha) & C = non-case \end{cases}$$

If $h_c$ denotes the constant historical force of infection in country $c$, then the probabilities $\pi_{bc}(\alpha)$ of having serostatus $b$ in country $c$ at age $\alpha$ are given by

$$\pi_{0c}(\alpha) = e^{-h_c\alpha}$$

then the likelihood of parameters $\theta$ is given by

$$\mathfrak{L}(\theta) = \prod_{i=1}^{N}\left(\mathbb{P}^*_{a_ib_ic_i}(C_i, t_S i, t_E i, \alpha_i)\pi_{b_ic_i}(\alpha_i)\right)$$

## Data augmentation

We have baseline immunity data for only around 10% of subjects, and the above likelihood requires the baseline serostatus of each trial participant. We employ data augmentation, in which the baseline serostatus of each participant outside the immunogenicity subset is treated as a parameter, to infer the immunological status of each participant with missing baseline serostatus. This has the advantage that fitted parameters are less dependent on initial assignment of baseline serostatus and can be considered as marginal distributions over possible values of baseline immunity. We use Gibbs sampling to calculate the conditional probability of seropositivity, given the current state of the parameter chain $\theta$ and the patient's age, trial arm, country and clinical outcome $C$.

Abbreviating and letting $\mathbb{P}(S_+|C; \theta)$ and $\mathbb{P}(S_-|C; \theta) = 1 - \mathbb{P}(S_+|C; \theta)$ respectively denote the probabilities of seropositivity and seronegativity at baseline given clinical outcome $C$, by Bayes' theorem we have

$$\mathbb{P}(S_+|C; \theta) = \frac{\mathbb{P}(C|S_+; \theta)}{\mathbb{P}(C; \theta)} = \frac{\mathbb{P}(C|S_+; \theta)}{\mathbb{P}(C|S_+; \theta)\mathbb{P}(S_+; \theta) + \mathbb{P}(C|S_-; \theta)\mathbb{P}(S_-; \theta)}$$

For example, a non-case of age $\alpha$ in country $c$ has the following probability of seropositivity at baseline

$$\mathbb{P}(S_+|\text{Non-case}; \theta) = \frac{\mathbb{P}_{a,1,c}(t_S, t_E, \alpha)}{\mathbb{P}_{a,1,c}(t_S, t_E, \alpha)(1 - \exp(-h_c\alpha)) + \mathbb{P}_{a,0,c}(t_S, t_E, \alpha)\exp(-h_c\alpha)}$$

for parameters $\theta$. Similarly, for a case of severity $D$ of age $\alpha$ in country $c$ we have

$$\mathbb{P}(S_+|\text{Case}; \theta) = \frac{\mathbb{Q}_{a,1,c,D}(t_S, t_E, \alpha)}{\mathbb{Q}_{a,1,c,D}(t_S, t_E, \alpha)(1 - \exp(-h_c\alpha)) + \mathbb{Q}_{a,0,c,D}(t_S, t_E, \alpha)\exp(-h_c\alpha)}$$

## Hazard ratios

If $\mathfrak{V}$ and $\mathfrak{C}$ are the sets of vaccinees and controls, respectively, then within any given stratum of interest $\mathfrak{S}$ (e.g. in a particular country, age or serostatus subset, or combination thereof), then posterior ratios of hazards of any disease severity and due to any serotype $HR(t^*)$ for all serotypes combined at time post-first dose $t^*$ are given by

$$HR_{\mathfrak{S}}(t*) = \frac{\sum_{i \in \mathfrak{S} \cap \mathfrak{B}} \lambda_{a_i b_i c_i}(t_{S_i} + t*, \alpha_i)}{\sum_{i \in \mathfrak{S} \cap \mathfrak{C}} \lambda_{a_i b_i c_i}(t_{S_i} + t*, \alpha_i)} \frac{|\mathfrak{S} \cap \mathfrak{C}|}{|\mathfrak{S} \cap \mathfrak{B}|}$$

where $|\mathfrak{S}|$ denotes the number of trial participants in stratum $\mathfrak{S}$.

For hazard ratios $HR(t*,d)$ of a particular disease severity $D$ and serotype $d$, we have

$$HR_{\mathfrak{S}}(t*, d, D) = \frac{\sum_{i \in \mathfrak{S} \cap \mathfrak{B}} \lambda_{a_i b_i c_i dD}(t_{S_i} + t*, \alpha_i)}{\sum_{i \in \mathfrak{S} \cap \mathfrak{C}} \lambda_{a_i b_i c_i dD}(t_{S_i} + t*, \alpha_i)} \frac{|\mathfrak{S} \cap \mathfrak{C}|}{|\mathfrak{S} \cap \mathfrak{B}|}$$

## Survival curves

For stratum $\mathfrak{S}$ as at $t*$ days post-first dose, posterior survival probabilities $\mathbb{P}_{\mathfrak{S}}(t*)$ are calculated as

$$\mathbb{P}_{\mathfrak{S}}(t*) = \frac{\sum_{i \in \mathfrak{S}} \mathbb{P}_{a_i b_i c_i}(t_{S_i}, t_{S_i} + t*, \alpha_i)}{|\mathfrak{S}|}$$

and if $n_{\mathfrak{S}}(t*)$ denotes the number of cases in stratum $\mathfrak{S}$ that occurred within $t*$ days post-first dose, then the observed survival probabilities are given by

$$\mathbb{P}_{\mathfrak{S},\mathrm{Obs}}(t*) = 1 - \frac{n_{\mathfrak{S}}(t*)}{|\mathfrak{S}|}$$

## Attack rates

The period for which participants were under active or passive surveillance varies by patient. Therefore, we calculate attack rates using

$$AR(\mathrm{TrialPeriod}) = \frac{\sum_{i \in \mathfrak{S}} \mathbb{P}_{a_i b_i c_i}(t_{S_i}, t_{A_i}, \alpha_i) - \mathbb{P}_{a_i b_i c_i}(t_{S_i}, t_{B_i}, \alpha_i)}{\sum_{i \in \mathfrak{S}} t_{B_i} - t_{A_i}}$$

where $t_{A_i}$ and $t_{B_i}$ are the start and end times of the trial period for patient $i$ ($t_{S_i}$ is the start of follow-up and is arbitrary here). To compute attack rates for observed data, we use the same formula, but $\mathbb{P}_{a_i b_i c_i}(t_{S_i}, t, \alpha_i)$ takes value 0 if the patient $i$ is a case between $t_{S_i}$ and $t$, and 1 otherwise. We use exact binomial confidence intervals on aggregate observed survival probabilities. For predicted attack rates, we use 95% credible intervals of posterior samples.

## Model variants and fitting

Model fitting was performed using the Metropolis–Hastings algorithm for parameter inference and Gibbs sampling for data augmentation. Parameters fitted include the relative risks, vaccine-induced transient immunities (by baseline serostatus, serotype and age) and their durations. For each country-specific baseline hazard, the logged knots of the spline are the fitted parameters, which explicitly determine all values of the baseline hazard. Prior distributions for parameters and augmented data were uniform (*Supplementary file 1*), and proposal distributions were normal. Each model variant was run for 1,100,000 iterations with a burn-in period of 100,000, storing 1 in every 100 iterations as posterior samples. Convergence was assessed visually. The model was coded in C++ using OpenMP (*Dagum and Menon, 1998*), and results were analysed in R 3.6.1 (*R Development Core Team, 2019*). All model code is available at https://github.com/dlaydon/DengVaxSurvival (*Laydon, 2021*; copy archived at swh:1:rev:d4964b7240312a371b2767533099643c59025dbf).

We consider alternative model variants that do not incorporate explicit serotype or age effects (*Figure 2B–D*, *Table 1*), and also a variant without vaccine-induced immune priming. Model fit is assessed both visually and using the Bayesian Information Criterion (BIC) (*Bhat and Kumar, 2010*) and the Widely Applicable Bayesian Information Criterion (WBIC) (*Watanabe, 2013*). For the WBIC, model variants with the highest values are to be preferred, in contrast to the BIC where model variants with the lowest values are preferred.

**Table 1.** Parameter values by model variant.

[A, B, C, D] refer to panels A, B, C and D of *Figure 2*. WBIC: Widely-Applicable Bayesian Information Criterion.

| Model variant | | | Main model [A] | Age effects only[B] | Serotype effects only[C] | Simplest model[D] |
|---|---|---|---|---|---|---|
| WBIC | | | −28627.1 | −29065.33 | −28591.5 | −29018.16 |
| Relative risks | Trial phase | No. prior infections | | | | |
| | Active ($K_{0,0}$) | 0 | 0.7 (0.36, 0.98) | 0.53 (0.26, 0.9) | 0.52 (0.28, 0.83) | 0.43 (0.24, 0.78) |
| | Active ($K_{0,1}$) | 1 (baseline) | 1 (1, 1) | 1 (1, 1) | 1 (1, 1) | 1 (1, 1) |
| | Active ($K_{0,2}$) | two or 3 | 0.31 (0.14, 0.63) | 0.32 (0.14, 0.65) | 0.2 (0.1, 0.4) | 0.21 (0.089, 0.45) |
| | Passive ($K_{1,0}$) | 0 | 0.053 (0.034, 0.078) | 0.054 (0.035, 0.078) | 0.051 (0.033, 0.073) | 0.052 (0.034, 0.074) |
| | Passive ($K_{1,1}$) | 1 | 0.21 (0.14, 0.31) | 0.22 (0.14, 0.31) | 0.2 (0.13, 0.29) | 0.21 (0.14, 0.3) |
| | Passive ($K_{2,1}$) | 2 or 3 | 0.053 (0.034, 0.078) | 0.054 (0.035, 0.078) | 0.051 (0.033, 0.073) | 0.052 (0.034, 0.074) |
| Seronegative transient immunity | Any serotype | Any age | - | - | - | 0.64 (0.34, 0.81) |
| | | 2–5 years | - | 0.48 (0.085, 0.77) | - | - |
| | | 6–11 years | - | 0.65 (0.36, 0.85) | - | - |
| | | 12–16 years | - | 0.61 (0.25, 0.85) | - | - |
| | Serotype 1 | Any age | - | - | 0.47 (0.12, 0.74) | - |
| | | 2–5 years | 0.16 (–0.34, 0.61) | - | - | - |
| | | 6–11 years | 0.39 (–0.0023, 0.73) | - | - | - |
| | | 12–16 years | 0.4 (0.022, 0.73) | - | - | - |
| | Serotype 2 | Any age | - | - | 0.29 (0.0095, 0.62) | - |
| | | 2–5 years | −0.11 (–0.72, 0.46) | - | - | - |
| | | 6–11 years | 0.12 (–0.38, 0.57) | - | - | - |
| | | 12–16 years | 0.14 (–0.39, 0.6) | - | - | - |
| | Serotype 3 | Any age | - | - | 0.69 (0.29, 0.96) | - |
| | | 2–5 years | 0.43 (–0.17, 0.86) | - | - | - |
| | | 6–11 years | 0.65 (0.15, 0.96) | - | - | - |
| | | 12–16 years | 0.67 (0.17, 0.97) | - | - | - |
| | Serotype 4 | Any age | - | - | 0.78 (0.59, 0.92) | - |
| | | 2–5 years | 0.54 (0.11, 0.85) | - | - | - |
| | | 6–11 years | 0.76 (0.53, 0.93) | - | - | - |
| | | 12–16 years | 0.78 (0.48, 0.98) | - | - | - |

*Table 1 continued*

| Model variant | | | Main model [A] | Age effects only[B] | Serotype effects only[C] | Simplest model[D] |
|---|---|---|---|---|---|---|
| Seropositive transient immunity | Any serotype | Any age | - | - | - | 0.43 (0.041, 0.73) |
| | | 2–5 years | - | 0.23 (0.0089, 0.6) | - | - |
| | | 6–11 years | - | 0.49 (0.11, 0.74) | - | - |
| | | 12–16 years | - | 0.65 (0.37, 0.82) | - | - |
| | Serotype 1 | Any age | - | - | 0.29 (0.015, 0.64) | - |
| | | 2–5 years | 0.21 (0.011, 0.55) | - | - | - |
| | | 6–11 years | 0.42 (0.063, 0.7) | - | - | - |
| | | 12–16 years | 0.54 (0.22, 0.78) | - | - | - |
| | Serotype 2 | Any age | - | - | 0.4 (0.041, 0.75) | - |
| | | 2–5 years | 0.26 (0.015, 0.63) | - | - | - |
| | | 6–11 years | 0.47 (0.1, 0.76) | - | - | - |
| | | 12–16 years | 0.59 (0.28, 0.83) | - | - | - |
| | Serotype 3 | Any age | - | - | 0.54 (0.12, 0.84) | - |
| | | 2–5 years | 0.36 (0.06, 0.72) | - | - | - |
| | | 6–11 years | 0.57 (0.19, 0.83) | - | - | - |
| | | 12–16 years | 0.69 (0.39, 0.9) | - | - | - |
| | Serotype 4 | Any age | - | - | 0.79 (0.44, 0.98) | - |
| | | 2–5 years | 0.52 (0.2, 0.86) | - | - | - |
| | | 6–11 years | 0.73 (0.37, 0.96) | - | - | - |
| | | 12–16 years | 0.85 (0.56, 0.99) | - | - | - |
| Transient-immunity duration | | Seronegative ($\tau_0$) | 4.5 (1, 9.6) | 5 (1.1, 9.7) | 4.1 (0.97, 9.4) | 4.9 (1.2, 9.6) |
| | | Seropositive ($\tau_1$) | 11 (2.3, 20) | 12 (2.9, 20) | 9.5 (1.2, 19) | 10 (1.3, 20) |
| Age-specific hazard multiplier $Z(\alpha)$ | | 2–5 years (baseline) | 1 (1,1) | 1 (1, 1) | - | - |
| | | 6–11 years | 1.2 (0.91, 1.5) | 1.1 (0.88, 1.4) | - | - |
| | | 12–16 years | 1.1 (0.79, 1.5) | 1 (0.76, 1.4) | - | - |
| Infection risk multiplier | | Seronegative ($M_0$, baseline) | 1 (1,1) | 1 (1, 1) | 1 (1,1) | 1 (1, 1) |
| | | Seropositive ($M_1$) | 0.77 (0.43, 0.99) | 0.61 (0.31, 0.96) | 0.69 (0.4, 0.97) | 0.57 (0.33, 0.94) |

## Results

### Trial data

*Figure 3* shows the proportion of participants with virologically detected dengue by trial, age group, trial phase, serotype and disease severity. Both trials show a clear benefit of vaccine across each age group for the active phase (25 months post-first dose), where surveillance could detect both hospitalised and non-hospitalised disease. In the passive phase (next 11 months following active phase), there are considerably fewer cases, owing to its shorter duration and detection of only hospitalised cases. Further, the benefits of vaccination in the passive phase are less than the active, and 2–5-year-old vaccinees show a greatly increased risk of hospitalisation than controls. In both trials, a mix of infecting serotypes among cases was observed.

### Model outputs

Because we consider both transient immunity and the change in disease risk induced by vaccination, the vaccine's overall effect can be difficult to interpret using only parameter estimates. We therefore summarise model output using hazard ratios (vaccine/control). *Figure 4* shows estimated posterior

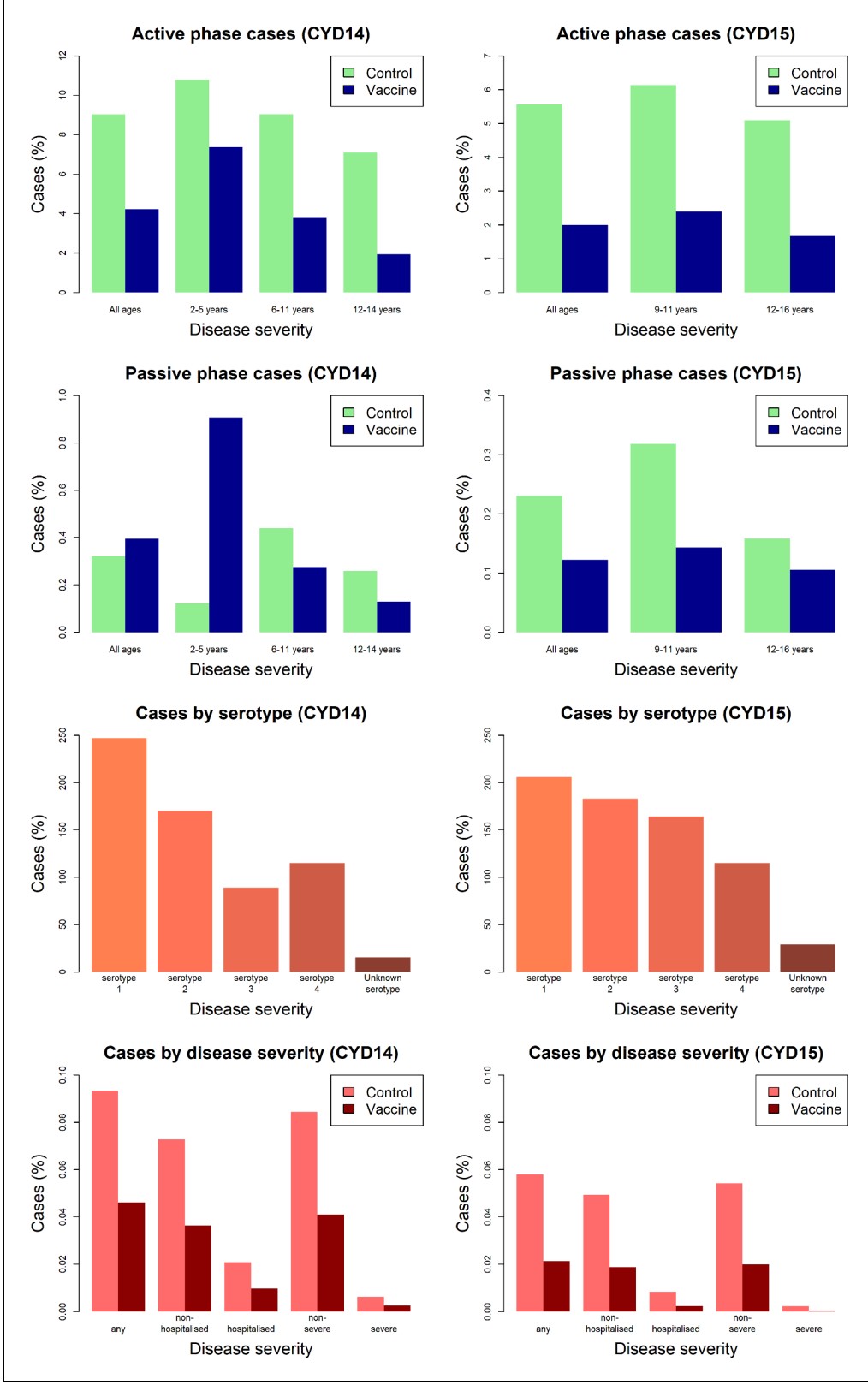

**Figure 3.** Summary of trial data.

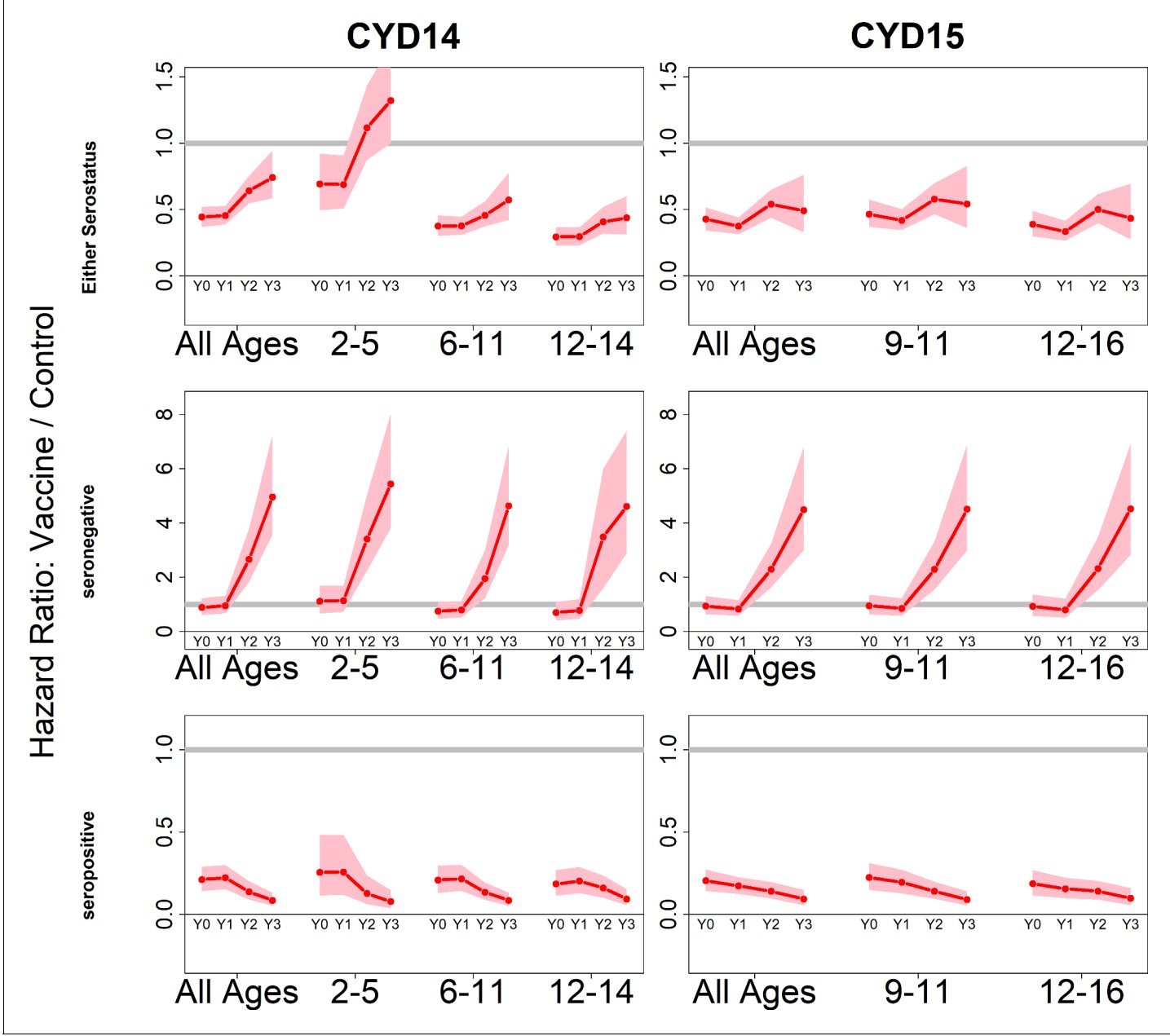

**Figure 4.** Posterior hazard ratios by trial, age group and baseline serostatus over time (main model). Each plot shows posterior hazard ratios (vaccine: control) for each age group at 0, 12, 25 and 36 months (0, 1, 2 or 3 years) of follow-up. Hazard ratios consider symptomatic disease regardless of serotype or hospitalisation status. Grey line indicates a ratio of 1, that is, no difference between the trial arms. Red line is the mean posterior estimate, and pink intervals represent 95% credible intervals of posterior samples. Rows show ratios when not broken down by serostatus (top row), for seronegatives (middle row) and seropositives (bottom row), for CYD14 (left column) and CYD15 (right column).

The online version of this article includes the following figure supplement(s) for figure 4:

**Figure supplement 1.** Posterior hazard ratio estimates by trial, age group, baseline serostatus and serotype over time (main model).

**Figure supplement 2.** Posterior hazard ratios by hospitalisation status.

**Figure supplement 3.** Posterior hazard ratios by serotype against non-hospitalised disease status.

**Figure supplement 4.** Posterior hazard ratios by serotype and hospitalised disease status.

hazard ratios for symptomatic disease (regardless of hospitalisation) by trial, serostatus and age group over time.

The vaccine has the greatest benefit for seropositive recipients: hazard ratios remain low throughout the active phase and the first year of passive phase, and mean posterior and 95% credible intervals are below 1 for all ages, indicating consistent benefit. The decrease in hazard ratios over time in each age group reflects the greater benefit to seropositives against hospitalised/severe disease as only this disease outcome is measured in the passive phase.

For seronegative vaccinees, hazard ratios are neither significantly positive nor negative during the first year. Ratios rise dramatically afterwards, reflecting low and short-lived immunity, combined with an almost sixfold long-term increase in disease risk, in both trials and in all age groups. Because the passive phase trial surveillance detected only hospitalised disease, this increase in hazard ratios refers to an increase in risk of hospitalisation, consistent with the 2017 NS1 data (*Sridhar et al., 2018*), to which our model was not fitted.

When unstratified by baseline serostatus, the vaccine is broadly beneficial, although hazard ratios rise over time, and are above 1 for 2–5 year olds in the first year of passive follow up. Decreasing hazard ratios with age reflect increasing seropositivity with age, and to a lesser extent the increase with age in transient immunity that we infer for seropositives. Low seroprevalence in 2–5 year olds is the driving factor of their increased risk, and this is consistent with the risk enhancement observed in the first year of long-term follow-up data. Hazard ratios are similar between trials for comparable age groups.

The estimated trends with age, serostatus and time hold when broken down by serotype (*Figure 4—figure supplement 1*), although net vaccine efficacy varies by serotype. Vaccination respectively offers the least and greatest benefit to serotypes 2 and 4. Hazard ratios are higher for serotype 2, although the vaccine remains beneficial in seropositives. For serotypes 3 and 4, hazard ratios are lower, and vaccination provides some initial protection even in seronegatives, although again these ratios rise over time. Risk enhancement of 2–5-year-old vaccinees is much higher for serotypes 1 and 2 than for 3 and 4.

*Table 1* shows parameter estimates for our main model. We infer relative risks by prior infection under both active and passive surveillance, and define $K_{i,D}$ as the relative risk of disease of type $D$ (in this instance referring to either the active or passive phase) given $i$ previous infections. Disease risk in active phase secondary infections is our baseline (i.e. $K_{1,0}$: = 1).

We estimate the relative risk parameters $K_{0,0}$ and $K_{2,0}$ to be 0.7 (0.36–0.98) and 0.31 (0.14–0.63), respectively. That is, among unvaccinated individuals, primary infection is 70% as likely, and tertiary/quaternary infection is 31% as likely, to cause symptomatic disease as secondary infection. However, considering vaccination as a silent infection, seronegative vaccinees increase their long-term risk of symptomatic disease by a factor of $K_{1,0}/K_{0,0}$ = 1.5 (1.0–2.7), whereas vaccinees with a single prior infection multiply their long-term disease risk by a factor of $K_{2,0}/K_{1,0}$ = 0.31 (0.14–0.63), transient immunity notwithstanding.

We estimate the proportions of symptomatic primary, secondary and tertiary/quaternary infections that require hospitalisation (i.e. parameters $K_{0,1}$, $K_{1,1}$ and $K_{2,1}$) to be 0.053 (0.034–0.078), 0.21 (0.14–0.31) and 0.053 (0.034–0.078), respectively. Therefore, in seronegatives, vaccination increases long-term risk of hospitalisation by a factor of $K_{1,0} K_{1,1}/K_{0,0} K_{0,1}$ = 6.1 (4.1–11), whereas for those with a single prior infection hospitalisation risk is multiplied by a factor of $K_{2,0} K_{2,1}/K_{1,0} K_{1,1}$ = 0.078 (0.036–0.16), again without considering transient immunity. Because disease risk in tertiary and quaternary infection is assumed to be equal (i.e. $K_{2,1} = K_{3,1}$ and $K_{2,2} = K_{3,2}$), multitypic seropositives (those with more than one previous infection) are unaffected by this mechanism of vaccine action.

Estimates of transient immunity by age, serostatus and serotype are shown in *Figure 5*. For every serotype, seronegative transient immunity estimates do not vary for older age groups but are lower for 2–5 year olds. They are lowest for serotype 2 (with negative mean estimates of −11% (−72–46%) for 2–5 year olds) and relatively high for serotypes 3 and 4. For seropositives, while estimates are again higher for serotypes 3 and 4, they vary less by serotype but more by age, with immunity being lower for younger than older children. Our results imply that transient immunity varies by age, independently of serostatus, although more so in seropositives (*Figure 5*, *Table 1*).

We could not precisely infer durations of transient immunity. Mean posterior estimates of seronegative and seropositive transient immunity duration are 4.5 (1–9.6) and 11 (2.3–20) years, respectively. However, a closer look at parameter posterior distributions (*Figure 5—figure supplement 1*,

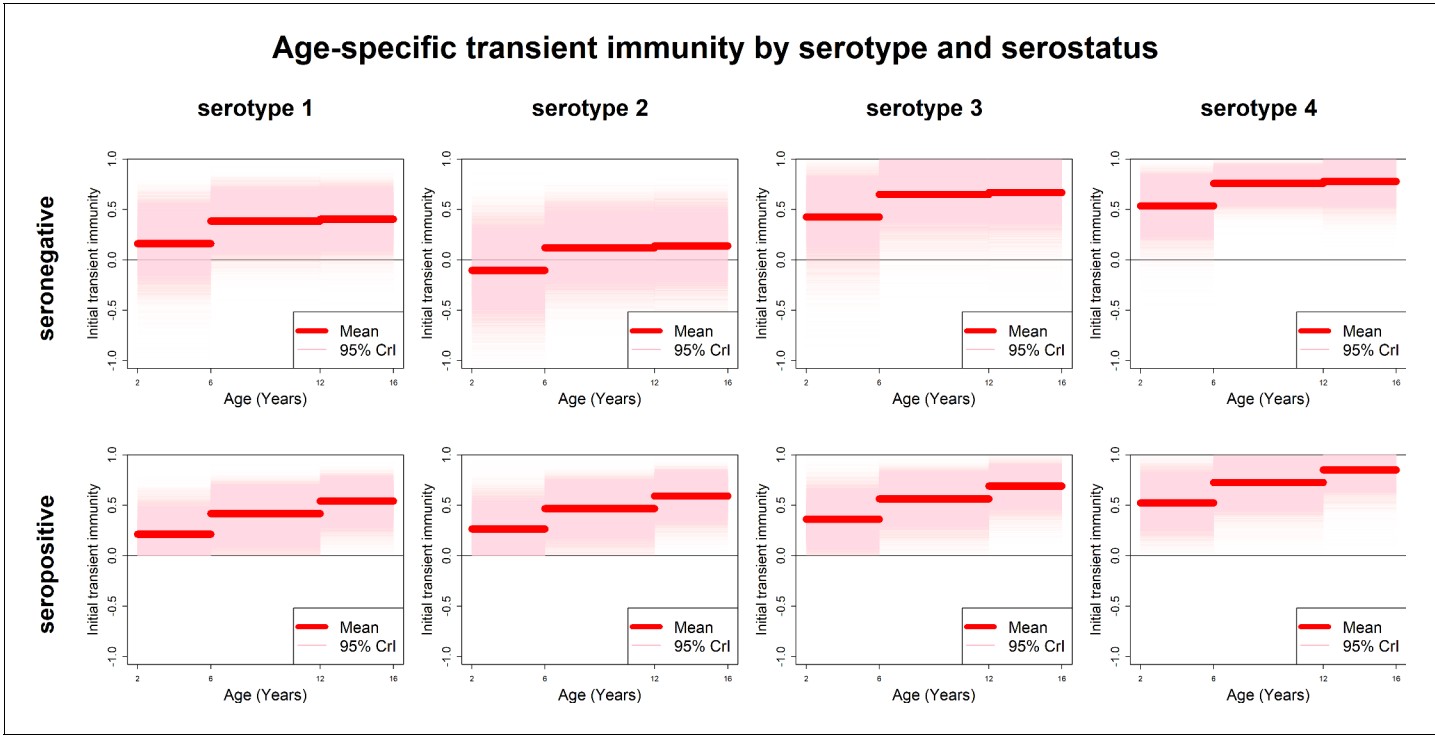

**Figure 5.** Age-specific transient immunity estimates (main model). Profiles of age-group-specific transient immunity by serostatus for serotypes 1–4. Mean posterior estimates are shown in red, with 95% credible intervals surrounding.

The online version of this article includes the following figure supplement(s) for figure 5:

**Figure supplement 1.** Transient immunity duration posteriors.

**Figure supplement 2.** Posterior chains of serotype proportions by country.

**Figure supplement 3.** Observed and predicted age-specific seroprevalence by country and trial.

**Figure supplement 4.** Age-specific transient immunity estimates (age effects model without serotype effects).

**Figure supplement 5.** Age-specific transient immunity durations (model with fitted age-specific transient immunity durations).

top row) is informative: in seropositives, approximately equal weight is given to longer durations of between 5 and 20 years (85% of posterior mass is above 5 years), whereas in seronegatives shorter durations are more likely (modal estimate is approximately 2 years and 62% of posterior mass is below 5 years).

If seropositive transient immunity were zero (or alternatively if the duration of transient immunity in seropositives was very short), then vaccination would only prime immunity and only individuals with pre-existing monotypic immunity would benefit from vaccination. Instead we estimate positive values for transient immunity for each age group and serotype. Further, model fits that fix seropositive transient immunity at zero do not reproduce the trial data. Therefore, for seropositives, to the extent that transient immunity is long-lived, vaccination confers benefit beyond that of priming immunity and consequent reduction of disease risk to that associated with natural tertiary and quaternary infection. Hence individuals with pre-existing multitypic immunity are also predicted to benefit from vaccination, with the caveat that we were not able to test a model in which transient immunity only applied to those with monotypic immunity. Conversely, in seronegatives, any positive benefit that mitigates long-term increase in disease risk is short-lived.

We find that the age group-specific multiplier of the baseline hazard increases with age for 6–11 year olds (1.2, 0.91–1.5), but then decreases for the 12–16 age group (1.1, 0.79–1.5) to be no different from the 2–5-year-old age group in the mean posterior estimates. While both credible intervals encompass 1 (indicating no difference), model runs that omit this age-specific hazard multiplier give a noticeably inferior visual fit (Appendix 1). Regardless of vaccination, we find that seropositives are 0.77 (0.43–0.99) times as likely to be infected as seronegatives due to their immunity to at least one serotype.

Estimates of the serotype proportions by country are shown in *Figure 5—figure supplement 2*, showing substantial heterogeneity between countries in their serotype distributions (e.g. Puerto Rico and Brazil's cases are almost exclusively comprised of serotypes 1 and 4, respectively).

## Model fits

Observed Kaplan–Meier curves demonstrate a clear overall benefit of vaccination. Over the combined active and first year passive phase, controls acquire symptomatic disease more than vaccinees in every country and age group. In both trials, vaccine efficacy wanes over time, and the slopes in the Kaplan–Meier curves become more equal *(Figure 6)*. The active phase lasted ~25 months (760 days), after which the slopes of the curves in each trial arm level off because only hospitalisations were detected. The model was fitted to the combined data from CYD14 and CYD15, and reproduces the observed Kaplan–Meier curves well across countries and age groups *(Figure 6)*.

Observed attack rates varied widely by country, trial arm, trial phase and age group *(Figure 7, Figure 7—figure supplements 1* and *2)*. Attack rates were generally higher in CYD14 than CYD15, and they decrease with age in both arms. Our main model captures this variation well, and the mean predicted attack rates fall within the confidence intervals of the observed attack rates in every age group, country, trial arm and trial phase. Importantly, the mean estimates reproduce the increased hospitalisation among 2–5-year-old vaccinees observed in the CYD14 trial *(Figure 7)*.

*Figure 7—figure supplements 3* and *4* show attack rates outputted from the immunogenicity subset only, where model predictions remain within the confidence intervals of observed attack rates. Attack rates again decrease with age in seropositives but not seronegatives, likely because of immunity to previously encountered serotypes *(Figure 7—figure supplement 4)*. For seronegative vaccinees, increased disease risk in the passive phase is predicted for all age groups in both trials, and not only 2–5 year olds in CYD14. Conversely, predicted seropositive attack rates are lower for vaccinees than controls across both trials. Our estimates well reproduce the data that is not overly sparse and again largely predict the 2017 NS1 data *(Sridhar et al., 2018)*. The observed distribution of seroprevalence by age in the immunogenicity subset is well mirrored in the augmented data *(Figure 5—figure supplement 3)*.

## Severity analysis

Our default interpretation of relative risk parameters $K_{i,D}$ distinguishes between differences in case detection under active and passive surveillance. However, we can also interpret these parameters to distinguish between non-hospitalised and hospitalised disease (or alternatively between non-severe and severe disease) (see Materials and methods).

We find that hazard ratios are consistent with our default interpretation which does not consider disease severity. However, here it is easier to distinguish between the vaccine's temporal effects and its differing efficacy against hospitalised or severe disease. For seropositives, temporal effects are minimal, and vaccination confers high and long-lasting protection against disease, and more so against hospitalised disease. In seronegatives, against non-hospitalised disease, hazard ratios again show only minor (and sometimes non-significant) initial protection, but they do not rise to the same dramatic extent. However, against hospitalised disease there is immediate risk enhancement that substantially worsens over time *(Figure 4—figure supplement 2)*. In summary, the differences between seronegative and seropositive efficacy are greater against hospitalised disease than against febrile disease.

These trends hold when broken down by serotype *(Figure 4—figure supplements 3* and *4)*. Protection is greatest against serotype 4 and lowest against serotype 2. The rate at which vaccination enhances hospitalisation risk in seronegatives varies by serotype: against serotype 2, vaccination increases hospitalisation risk immediately after vaccination and only rises slowly during the following three years (to approximately sixfold); whereas for serotypes 3 and 4 the eventual risk enhancement is lower (approximately fourfold) but follows a delay *(Figure 4—figure supplement 4)*. Hazard ratios are almost identical when considering severe and non-severe disease.

Age-, serotype- and serostatus-specific transient immunity estimates are similar to our default interpretation when considering hospitalised or severe disease (not shown). We allow the proportions of symptomatic disease requiring hospitalisation to differ between trials to reflect non-standardised criteria between countries. Among secondary infection, proportions differ slightly between the

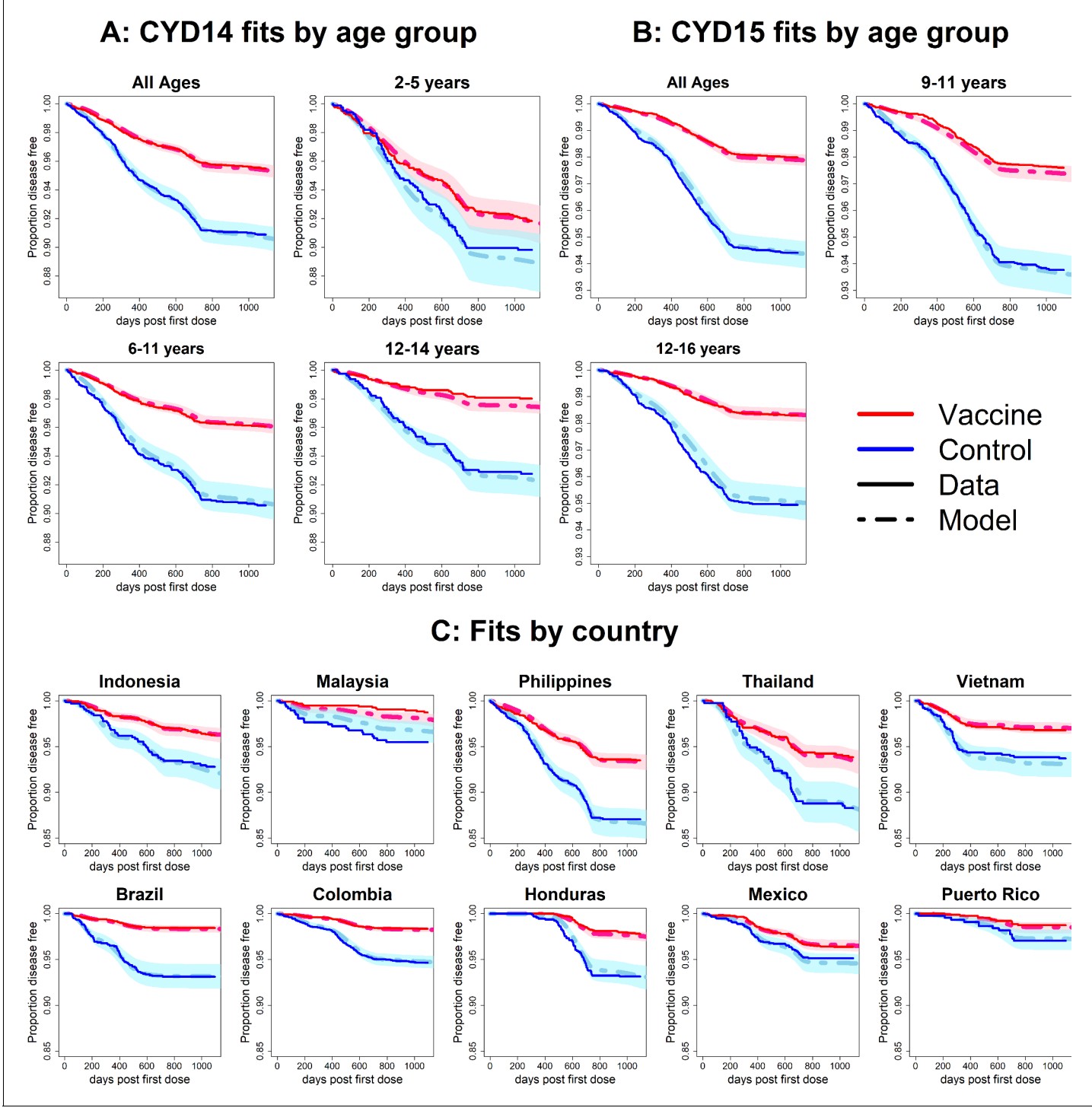

**Figure 6.** Fits to observed Kaplan–Meier curves by trial, age group and country (main model). In each plot, vaccine and control survival probabilities are plotted against days post-first dose (vaccine or placebo). Dark lines denote observed curves, dashed lines are the mean posterior estimates, with 95% credible intervals around.

The online version of this article includes the following figure supplement(s) for figure 6:

**Figure supplement 1.** Fits to observed Kaplan–Meier curves by trial and disease severity.

**Figure supplement 2.** Fits to observed Kaplan–Meier curves by trial and hospitalisation status.

**Figure supplement 3.** Fits to observed Kaplan–Meier curves by trial, age group and country (simplest model without age or serotype effects).

**Figure supplement 4.** Fits to observed Kaplan–Meier curves by trial, age group and country (age effects model without serotype effects).

**Figure supplement 5.** Fits to observed Kaplan–Meier curves by trial, age group and country (serotype effects model without age effects).

*Figure 6 continued on next page*

CYD14 and CYD15 trials at 0.30 (0.23–0.33) and 0.16 (0.12–0.21), respectively (values for primary and tertiary/quaternary infection are determined by fixed ratios). Relative risks of severe disease are considerably lower than hospitalised disease at 0.012 (0.0087–0.016), 0.047 (0.035–0.063) and 0.012 (0.0087–0.016) for primary, secondary and tertiary/quaternary infection.

When distinguishing between severe and non-severe disease, we reproduce observed survival curves at trial level for non-severe disease, but fits are less good for severe disease (*Figure 6—figure supplement 1*). This is due firstly to limited data (non-severe cases outnumber severe cases by 1223 to 58), and secondly because we do not allow transient immunity to vary by disease severity. When we instead consider hospitalisation *(Figure 6—figure supplement 2)*, fits to survival curves are good regardless of hospitalisation status. In both scenarios, model fits to 'either' disease severity (e.g. surviving both severe and non-severe disease) closely resemble those of our default interpretation, where disease severity is not considered. Attack rates in each disease category are relatively well fitted, although passive phase attack rates are less well fitted for severe or hospitalised disease (non-hospitalised febrile disease is not detected by passive surveillance) (*Figure 7—figure supplements 5* and *6*).

## Alternative model variants

We conducted a sensitivity analysis to examine whether more parsimonious models are sufficient to explain the complex trial data (Appendix 1). Broadly, it is necessary to include explicit age effects to reproduce the age distribution of cases and to include serotype effects to reproduce variation by country. While we could not precisely infer the duration of transient immunity, our analysis indicated that it is short-lived in seronegatives and long-lived in seropositives.

## Discussion

Our results provide a comprehensive profile of Sanofi-Pasteur's CYD-TDV vaccine (*Dengvaxia*). We investigated multiple mechanisms of vaccine action and analysed its dependence on serotype, baseline serostatus and age. We further examined efficacy by disease severity.

There was substantial heterogeneity in transient immunity by serotype and serostatus. Vaccine-induced protection against each serotype was higher in seropositive recipients than in seronegatives, and these findings were robust across model variants. The incorporation of serotype-specific transient immunity improved model fits to country breakdowns, but had little effect on fits to age breakdowns. Interestingly, transient immunity was found to increase with age in seropositives and to a lesser extent in seronegatives. While one mechanism of our model (change in relative risk through 'silent infection') separates seropositivity into monotypic and multitypic immunity, the other (conferral of transient immunity) does not, and so it is possible that the age trend in seropositives also reflects increasing transient immunity for multitypic immunes, although the slight age trend in seronegatives could not be explained this way. In general, heterogeneity between countries' Kaplan–Meier curves can be explained by serotype and seroprevalence, although these factors are insufficient to explain differences in vaccine efficacy by age, for which age-specific effects (independent of serostatus) are required.

In every model variant examined, vaccination substantially decreased disease risk in seropositives, but increased risk in seronegatives (particularly risk of hospitalised/severe disease). These findings are consistent with and largely predict the NS1 data and long-term follow-up data (*Sridhar et al., 2018*). We further found larger differences in efficacy between seropositives and seronegatives when considering hospitalised disease: benefit to seropositives and risk enhancement in seronegatives is greater than against febrile disease. Our findings here may be affected by the data used: the passive surveillance in the first year of hospital follow-up does not detect non-hospitalised disease.

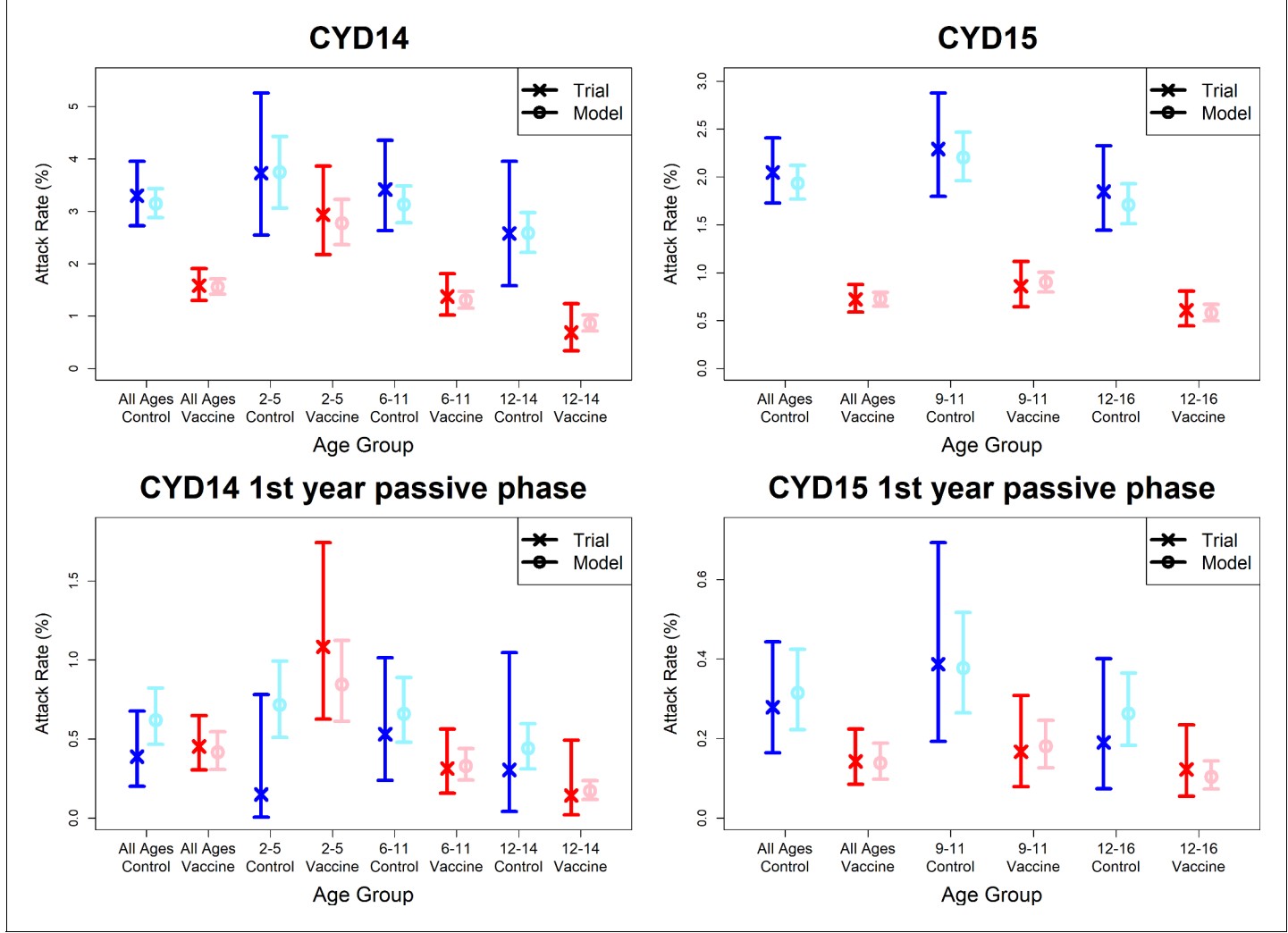

**Figure 7.** Model fits to observed attack rates at trial level for active and first year passive phase. Observed and predicted age group-specific attack rates are shown for CYD14 (left column) and CYD15 (right column), for first three years follow-up (active phase and first year hospital phase combined), and for the first year of follow-up only. Blue and red denote observed attack rates for control and vaccine groups, while light blue and pink denote model predictions for control and vaccine groups. Confidence intervals for observed attack rates are calculated using exact binomial confidence intervals, whereas the uncertainty around predicted rates are 95% posterior sample credible intervals.

The online version of this article includes the following figure supplement(s) for figure 7:

**Figure supplement 1.** Model fits to observed attack rates by country.
**Figure supplement 2.** Model fits to observed first year passive phase attack rates by country.
**Figure supplement 3.** Model fits to observed immunogenicity subset attack rates.
**Figure supplement 4.** Model fits to observed immunogenicity subset attack rates from first year passive phase.
**Figure supplement 5.** Model fits to observed attack rates by trial, hospitalisation status and trial phase.
**Figure supplement 6.** Model fits to observed attack rates by trial, disease severity and trial phase.
**Figure supplement 7.** Model fits to observed attack rates at trial level for active and first year passive phase (model without immune priming).
**Figure supplement 8.** Model fits to observed attack rates at trial level for active and first year passive phase (model with fitted age-specific transient immunity durations).

> While our analysis demonstrates that serostatus is the dominant factor in efficacy, the data do not allow us to consider the order of infecting serotype, which recent work suggests affects disease risk (*Aguas et al., 2019*) and therefore perhaps efficacy also. While we have analysed the vaccine's effect against different serotypes and different severities of disease, we have not analysed efficacy against infection (*Olivera-Botello et al., 2016*). It would be helpful to examine the degree to which efficacy

is dependent upon antibody titres (*Salje et al., 2018*; *Katzelnick et al., 2017*) as opposed to a binary serostatus. The latter may determine whether vaccine immune priming is age dependent.

Our model does not disaggregate seropositives into monotypic or multitypic immunity, and we were unable to test models whereby transient immunity applies only to multitypic seropositives. The use of antibody titres could be informative, although we are ultimately limited by the small size of the immunogenicity subset. Additionally, we do not model either serotype-specific natural immunity or transient immunity that arises from natural infection.

Our model estimates an enhancement of risk for seronegative vaccinees for every serotype (although more so for serotypes 1 and 2 than for serotypes 3 and 4), whereas previous work indicated the vaccine's better performance against serotype 4 (*Sridhar et al., 2018*). This is likely due to the fact that we do not consider serotype-specific relative risks (and therefore serotype-specific changes in relative risk induced by 'silent infection' vaccination). While we attempted previously to resolve this issue, there is insufficient power to resolve these parameters, particularly for the passive phase or hospitalised disease. Further, serotype-specific transient immunity durations would likely have diminished or altogether removed the predicted risk enhancement for serotype 4. Again though, there is insufficient power to resolve such serotype effects, particularly seeing as transient immunity durations by serostatus were not precisely inferred.

Current WHO guidance recommends serological testing of potential vaccine recipients before vaccination and only vaccinating seropositives (*Dengue vaccine, 2019*). Age targeting of vaccination is therefore important: too young an age, and most of those tested will be seronegative, too old and most will have already experienced secondary dengue infection.

Distinguishing between monotypic and multitypic infection is not usually possible in clinical practice. However, our results suggest that all seropositives are likely to benefit from vaccination, and further that vaccinating them will be more beneficial than merely boosting their immunity to that of someone with two previous natural infections. Importantly, this means it may be more beneficial to vaccinate multitypic seropositives than other models have predicted (*Flasche et al., 2016*), at least to the extent that seropositive transient immunity is long-lived and acts in both monotypic and multitypic seropositive vaccine recipients.

High-resolution maps of dengue seropositivity are now available, and alongside improved rapid diagnostic tests and 'screen-then-vaccinate' programmes (*Flasche and Smith, 2019*), optimal deployment of the vaccine could reduce the increasing worldwide burden of dengue disease by as much as 30% (*Cattarino et al., 2020*). Therefore, targeting only seropositive recipients with this vaccine is an increasingly viable public health strategy.

## Acknowledgements

DJL, GNG, and NMF acknowledge grant funding from the Bill & Melinda Gates Foundation (grant OPP1092240). DJL, ID, WRH, GNG and NMF acknowledge joint centre funding from the UK Medical Research Council and Department for International Development (grant MR/R015600/1). DJL, GNG and NMF acknowledge funding from Vaccine Efficacy Evaluation for Priority Emerging Diseases (VEEPED) grant (ref. NIHR: PR-OD-1017–20002) from the National Institute for Health Research. ID acknowledges research funidng from a Sir Hentry Dale Fellowship funded by the Royal Society and Wellcome Trust (grant 21349/Z/18/Z). Views expressed do not necessarily represent those of the funders. DJL, ID and NMF have advised Sanofi-Pasteur Ltd., without payment on the implications that this work has on the use of their vaccine. We thank Nick Jackson and Jean-Sebastien Persico at Sanofi-Pasteur, and Ben Lambert, Anne Cori and Natsuko Imai at Imperial College London for useful discussions. We further thank our reviewers and editors for useful comments. All clinical trial data used for these models are publicly available in the original publications (cited), and model code is available at https://github.com/dlaydon/DengVaxSurvival. *Role of the funding source:* The funder of the study had no role in study design, data collection, data analysis, data interpretation or writing of the report. The corresponding author had full access to all the data in the study and had final responsibility for the decision to submit for publication.

## Additional information

### Competing interests

Laurent Coudeville: Laurent Coudeville is employed by Sanofi-Pasteur. The other authors declare that no competing interests exist.

### Funding

| Funder | Grant reference number | Author |
|---|---|---|
| Bill & Melinda Gates Foundation | OPP1092240 | Daniel J Laydon<br>Gemma Nedjati-Gilani<br>Neil M Ferguson |
| National Institute for Health Research | NIHR: PR-OD-1017-20002 | Daniel J Laydon<br>Gemma Nedjati-Gilani<br>Neil M Ferguson |
| Medical Research Council | MR/R015600/1 | Daniel J Laydon<br>Ilaria Dorigatti<br>Wes R Hinsley<br>Gemma Nedjati-Gilani<br>Neil M Ferguson |
| Royal Society | Sir Henry Dale Fellowship grant 213494/Z/18/Z | Ilaria Dorigatti |
| Wellcome Trust | Sir Henry Dale Fellowship grant 213494/Z/18/Z | Ilaria Dorigatti |

The funders had no role in study design, data collection and interpretation, or the decision to submit the work for publication.

### Author contributions

Daniel J Laydon, Conceptualization, Software, Formal analysis, Validation, Investigation, Visualization, Methodology, Writing - original draft, Writing - review and editing; Ilaria Dorigatti, Formal analysis, Validation, Investigation, Methodology, Writing - review and editing; Wes R Hinsley, Software, Methodology, Writing - review and editing; Gemma Nedjati-Gilani, Software, Formal analysis, Investigation, Methodology, Writing - review and editing; Laurent Coudeville, Data curation, Validation, Writing - review and editing; Neil M Ferguson, Conceptualization, Software, Formal analysis, Supervision, Funding acquisition, Validation, Investigation, Visualization, Methodology, Writing - review and editing

### Author ORCIDs

Daniel J Laydon [ORCID] https://orcid.org/0000-0003-4270-3321
Ilaria Dorigatti [ORCID] http://orcid.org/0000-0001-9959-0706

### Decision letter and Author response

Decision letter https://doi.org/10.7554/eLife.65131.sa1
Author response https://doi.org/10.7554/eLife.65131.sa2

## Additional files

### Supplementary files

- Supplementary file 1. Model glossary / parameter table.

- Transparent reporting form

### Data availability

Qualified researchers may request access to patient level data and related study documents including the clinical study report, study protocol with any amendments, blank case report form, statistical

analysis plan, and dataset specifications. Patient level data will be anonymized and study documents will be redacted to protect the privacy of trial participants. Further details on Sanofi's data sharing criteria, eligible studies, and process for requesting access can be found at: https://www.clinicalstu-dydatarequest.com. Additional details of the trial designs and data can be found in Sridhar et al (NEJM 2018). All model code is available at https://github.com/dlaydon/DengVaxSurvival (copy archived at https://archive.softwareheritage.org/swh:1:rev:d4964b7240312a371b2767533099643c59025dbf), which is linked to in the manuscript. This repository also contains simulated data, generated to closely match the trial data, giving comparable case numbers across strata. When our model is fitted to the simulated data, the resulting parameter estimates closely approximate the results presented in this analysis.

The following datasets were generated:

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

## Appendix 1

### Simplest model without age or serotype effects

When we omit serotype and age effects (infection history notwithstanding) *(Figure 2D)*, the hazard is given by

$$\lambda_{abcD}(t,\alpha) = \lambda_c(t)M_b R_{abcD}(\alpha)\big(1 - \delta_{a,Vac}I_b^*(t,t_F)\big).$$

Here age is only considered alongside historical transmission intensity to determine the contributions of monotypic and multitypic infection to seropositive relative risks.

This reduced model fits observed Kaplan–Meier curves well at trial level, although age group breakdowns are more variable: vaccine efficacy is respectively overestimated and underestimated the 2–5 and 12–14 CYD-14 age groups *(Figure 6—figure supplement 3)*. This is likely due to an uneven age distribution in CYD-14: age groups 2–5 and 12–14 constitute 24.2% and 22.7% of CYD-14 (or 7.9% and 7.5% of combined CYD-14 and CYD-15 data), whereas 6–11 years olds comprise 53.2% of the CYD14 data (17.5% of combined data). CYD-15 age breakdowns are better *(Figure 6—figure supplement 3)*, attributable to its larger size and narrower age range, although efficacy is still slightly underestimated in the highest age group.

Fits to individual countries are more variable *(Figure 6—figure supplement 3)*. Lower-quality fits in country and age group subsets are usually attributable to thinner data and specifically lower case numbers; however, this explanation is insufficient for Brazil, which has high case numbers, indicating country-specific effects that are not captured by reduced model. The estimates of serotype proportions *(Figure 5—figure supplement 2)* are instructive: Brazil and Mexico have vastly different serotype proportions (specifically serotypes with the highest and lowest vaccine-induced immunities, respectively), and therefore fits to these countries improve with the inclusion of serotype effects. In contrast, Indonesia, Thailand and Colombia have more equal serotype proportions, and thus their fits are not greatly improved. These results indicate that the effects of vaccination vary by age, independently of serostatus, and therefore that a more complex model is necessary to explain the trial data.

Relative risk parameters are lower but consistent with our main model, and initial seronegative transient immunity is 0.64 (0.34–0.74), whereas seropositive transient immunity is estimated to be 0.43 (0.041–0.73). Seronegative and seropositive durations are estimated at 4.9 (1.2–9.6) and 10 (1.4–20) years, respectively (and have similar posteriors to our main model), again indicating long-lasting benefit to seropositives beyond the effect of immune priming *(Table 1)*.

### Model with age effects only

Adding age-specific transient immunity and force of infection *(Figure 2B)* to the simplest model, we have the hazard

$$\lambda_{abcD}(t,\alpha) = \lambda_c(t)M_b Z(\alpha) R_{abcD}(\alpha)\big(1 - \delta_{a,Vac}I_b^*(\alpha,t,t_F)\big).$$

Fits in the CYD-14 2–5 and 12–14-year-old age groups are improved; however, fits to individual countries are largely the same as the simplest model *(Figure 6—figure supplement 4)*. Further, each measurement of model fit is inferior to our main model *(Table 1)*. We conclude that inclusion of age effects alone is insufficient to explain the trial data.

This model variant again shows an increasing trend with age in transient immunity for seropositives, but less so for seronegatives *(Figure 5—figure supplement 4, Table 1)*. Mean transient immunity estimates are consistent with those from our main model when averaged over serotypes *(Table 1)*.

### Model with serotype effects only

Adding serotype-specific transient immunity to the simplest model *(Figure 2C)* gives the following hazard:

$$\lambda_{abcdD}(t,\alpha) = \lambda_c(t)\rho_{cd}M_b R_{abcD}(\alpha)\big(1 - \delta_{a,Vac}I_{bd}^*(t,t_F)\big)$$

where again age is only present in seropositive relative risk. While survival curves of individual countries are well fitted, age-stratified survival curves are fitted no better than the simplest model

and qualitatively worse than the model with age and serotype effects (*Figure 6—figure supplement 5*), and so we choose the latter model as our default. Vaccine-induced estimates from the model with only serotype effects are consistent with those from our main model when averaged over age groups (*Table 1*).

While these simpler model variants are more parsimonious, they do not explain the trial data as well as our main model. Age-specific variation in transient immunity and the force of infection are required to fit the age distribution of cases well, and serotype-specific transient immunities are necessary to fit observed attack rates by country.

### Model without age-specific force of infection

Because our estimates of the age-specific force of infection multiplier were relatively imprecise and had credible intervals that spanned 1 (indicating no difference between age groups), we investigated a model that removed these effects, with the following hazard

$$\lambda_{abcdD}(t,\alpha) = \lambda_c(t)\rho_{cd}M_b R_{abcD}(\alpha)\left(1 - \delta_{a,Vac}I^*_{bd}(\alpha,t,t_F)\right)$$

Transient immunity and relative risk parameters are essentially unaffected by this omission; however, the age breakdowns in CYD-14 are adversely affected in 2–5 year olds and 6–11 year olds (*Figure 6—figure supplement 6*).

### Model without immune priming

Our main model considers vaccination to prime host immunity akin to a natural disease-free infection, and this must be considered alongside estimates of vaccine-induced transient immunity. Here we consider a model without immune priming, and so relative risk parameters do not vary by trial arm, giving

$$R_{bc0}(\alpha) = \begin{cases} K_{0,0} & b=0 \\ \varphi_{c0}(\alpha) & b=1 \end{cases},$$

$$R_{bc1}(\alpha) = \begin{cases} K_{0,0}K_{0,1} & b=0 \\ \varphi_{c0}(\alpha)\varphi_{c1}(\alpha) & b=1 \end{cases},$$

and

$$\lambda_{abcdD}(t,\alpha) = \lambda_c(t)\rho_{cd}M_b Z(\alpha)R_{bcD}(\alpha)\left(1 - \delta_{a,Vac}I^*_{bd}(\alpha,t,t_F)\right)$$

In this model variant the benefit of the vaccine is purely a function of transient immunity and its duration.

Fits to the Kaplan–Meier curves (*Figure 6—figure supplement 7*) and attack rates (*Figure 7—figure supplement 7*) over the entire trial duration appear similar to our main model. However, passive phase attack rates in the 2–5-year CYD-14 age group are not reproduced at all, and indeed here the vaccine is erroneously predicted to be beneficial. Further, each measure of model fit is poorer for this variant. We conclude that immune priming is required to properly account for the trial data, explain the vaccine's action mechanism and evaluate vaccine benefit and risk.

### Model with age-specific durations of transient immunity

We investigated another model variant where we allowed the duration of protection $\tau_b(\alpha)$ to vary with age, and so here

$$I^*_{bd}(\alpha,t,t_F) = \begin{cases} I_{bd}(\alpha)\exp(-(t-t_F)/\tau_b(\alpha)) & I_{bd}(\alpha)>0 \\ I_{bd}(\alpha) & \text{Otherwise} \end{cases}$$

where $\tau_b(\alpha)$ is a step function.

Fits to Kaplan–Meier curves and attack rates are almost indistinguishable from our main model (*Figure 6—figure supplement 8* and *Figure 7—figure supplement 8*). We find no age dependence in seropositive durations, but there is some indication that seronegative transient immunity may be

longer-lived in older children (*Figure 5—figure supplement 5*). As posteriors for durations are already imprecisely inferred when not age-specific, and since fits are largely unchanged, we do not allow duration to vary in our main model.

