## [Decision Letter]

**Acceptance summary:**

This paper presents a detailed, model-based, characterisation of the efficacy of the only licensed dengue vaccine as a function of baseline serostatus and age. The results reinforce the hypothesis of "vaccination as a silent infection" and demonstrate the need for targeted vaccination using rapid diagnostic tests. Although this finding is not novel per se, having a precise characterisation of the time-varying risk is important for determining vaccine utility and optimal implementation strategies.

**Decision letter after peer review:**

Thank you for submitting your article "Efficacy profile of the CYD-TDV dengue vaccine revealed by Bayesian survival analysis of individual-level Phase III data" for consideration by *eLife*. Your article has been reviewed by 2 peer reviewers, and the evaluation has been overseen by a Reviewing Editor and a Senior Editor. The following individual involved in review of your submission has agreed to reveal their identity: James A Watson (Reviewer #2).

Essential revisions:

These revisions either reflect a need to temper some of the claims made in the paper or to improve clarity of the work.

1) The authors assume in their models that immunity wanes in seronegatives (over 1 year) and is stable in seropositives (maintained over time). It is unclear why these assumptions were made and clarification is needed.

2) The authors should adjust the statement "vaccinating seropositives gives greater long-lasting immunity than two natural infections" to reflect the fact that immunity in seropositives may also wane over time in a way that cannot be accurately measured with the current data.

3) On lines 329-330 the authors state "Therefore it may be more beneficial to vaccinate multitypic seropositives than simpler models have predicted [11]." How do the authors know it is the multitypics that benefit as opposed to those with primary immunity before vaccination?

4) It would be helpful if the authors further discussed age-effects and their relationship with being multitypic immune.

5) Please address reviewer 1's concern that the results of this manuscript do not support the claim on lines 311 – 315. Please also address concerns about lines 316-320.

6) Could the authors comment on whether the fact that serotype effects are more similar than might be expected based on previous articles on these data might relate to the way serotype effects are modeled, or whether this constitutes a major difference from what has been shown previously?

7) Please consider reviewer 1's suggestions to improve readability by defining KiD in the main text and revising table 1.

8) Line 349 – Where does the value of 0.7 for seropositives come from?

9) Please explain the big K notation in the Results section.

10) Please add a parameter table to the methods section and add a section on the data.

11) Please justify choice of priors and/or consider sensitivity of results to choice of priors.

12) Please provide access to the model code. The statement "Model code is available from the authors" does not meet *eLife* requirements: "Regardless of whether authors use original data or are reusing data available from public repositories, they must provide program code, scripts for statistical packages, and other documentation sufficient to allow an informed researcher to precisely reproduce all published results."

*Reviewer #1:*

Laydon et al. have conducted an elegant analysis that provides a clear and comprehensive guide to the mechanism of difference of CYD-TDV for dengue virus seropositive vs. seronegative vaccine recipients. The paper includes relevant hypothesis and model schemes as well as figures that show differences between seropositive and seronegative vaccine recipients across a range of covariates. The authors demonstrate, in a clearer way than has been shown before, that CYD-TDV increases disease across all ages in both trials in seronegatives. The authors also show that the enhancing effect is stronger against hospitalized disease than febrile disease. Many of these results have already been presented previously in other papers analyzing the same data, but the modeling approach does provide a new perspective that adds to the story of the CYD-TDV vaccine.

Overall, the paper is clearly written and easy to follow, especially given the complexity of the model and subject matter. However, numerous claims are made in the abstract and the discussion that inaccurately reflect the results presented in this paper. In particular, major conclusions of the paper relate to the benefit of the vaccine for seropositive individuals (lines 18-19, 327-328) and an increasing effect of vaccination for seropositive individuals with age (lines 286-290, 329-330). There are potential issues with the modeling approach and how it relates to these conclusions. Further, a discussion about the long-term effects of the vaccine for seronegatives is speculative and problematic (311-320).

Comments for the authors:

1) Lines 18-19. "Vaccinating seropositives gives greater long-lasting immunity than two natural infections." The claim that fixed-time immunity induced by vaccination of seropositives is maintained at a high level over time is not well supported. The authors assume in their models that immunity wanes in seronegatives (over 1 year) and is stable in seropositives (maintained over time). First, it is unclear why these assumptions were made. The reference for these model choices does not seem to cover immune kinetics over this period of time (Clapham et al. 2016 PLOS Comp Bio). Second, the authors actually tested different assumptions about waning immunity in supplement (lines 667 – 694), which are not described in the main text. In the supplement, the authors demonstrate waning immunity out to 4.5 years (with wide confidence intervals) in both seronegative and seropositive vaccinated individuals. In both cases, the duration of waning is outside of the observational period of the study, and the authors write: "We conclude that three years of follow up data from each patient is insufficient to infer durations." Thus, the authors should adjust the statement "vaccinating seropositives gives greater long-lasting immunity than two natural infections" to reflect the fact that immunity in seropositives may also wane over time in a way that cannot be accurately measured with the current data.

2) Lines 329-330. "Therefore it may be more beneficial to vaccinate multitypic seropositives than simpler models have predicted [11]." How do the authors know it is the multitypics that benefit as opposed to those with primary immunity before vaccination? It would be helpful if the authors could further support this statement by discussing the results that relate to this point. This is especially important in relation to the discussion about dengue rapid diagnostic tests (e.g. lines 23-24), which have very low sensitivity for detecting monotypic DENV immunes.

3) Lines 286 -290. "Interestingly, fixed-time immunity was found to increase with age in seropositives. In general, heterogeneity between countries' Kaplan-Meier curves can be explained by serotype and seroprevalence, although these factors are insufficient to explain differences in vaccine efficacy by age, for which age-specific effects (independent of serostatus) are required." The conclusion about the benefit of the vaccine even to multitypic immunes seems to be derived from the observation that fixed-effect immunity induced by the vaccine in seropositive individuals increases with age. The authors observe large age-specific fixed-time immunity differences by serostatus (Figure 5), although it is not clear there is much difference across age groups in vaccine efficacy (Figure 4). While the authors state that age specific effects are independent of serostatus, the age effects only occur in seropositives and thus are most plausibly related to more prior infections in children of older ages. It would be helpful if the authors further discussed age-effects and their relationship being multitypic immune.

4) Lines 311 – 315. "In high transmission settings where children would ordinarily receive at least two natural infections, we predict that even seronegatives would eventually benefit from vaccination, as they would experience one high risk infection, and all subsequent infections would have a much lower risk, in contrast to unvaccinated individuals who would ordinarily experience one moderate and one high risk infection [Figure 1]." The results of this manuscript do not support this claim. While the authors may expect this to be true based on the model scheme (referenced as Figure 1), the results, as currently described, do not prove this point. Further, the caveats provided afterward do not justify inclusion of this paragraph in the discussion (Lines 316-320): "Important caveats include first that disease risk would increase in the short term. Second, and more seriously, a small subset of symptomatic infections are fatal, in which case vaccination could shorten life in seronegatives. Finally, while the eventual benefit to seronegatives is predicted by our model and the model fits well, this remains a prediction yet to be validated by empirical data."). This paragraph should be removed or significantly reworked because it is largely a question of medical ethics.

5) Figure 4. It is not clear why vaccine efficacy increases over time in the trial for vaccinated seropositives, even across all age groups. It does not seem to be explained by age effects, waning immunity, or serotype distributions. Is it related to differences in disease outcomes measured in the trial over time (e.g. Figure S8)?

6) It is surprising that the serotype effects are more similar than might be expected based on previous articles on these data, especially previous observations of greater efficacy against DENV4. Here the authors observe strong enhancement of DENV4 at later timepoints in the trial. Could the authors comment on whether this might relate to the way serotype effects are modeled, or whether this constitutes a major difference from what has been shown previously?

7) Table 1 and its explanation in the text are confusing. Instead of a traditional presentation of relative risk, which is described in the text, the table presents the probability of disease expected relative to secondary infections. One option would be to present relative risk estimates in table, and clearly indicate the reference groups in the table. Additionally, while KiD is clearly defined in the methods, it would be helpful to add one sentence to the main text to define KiD, given that the main text uses this notation in multiple paragraphs without explaining its interpretation.

*Reviewer #2:*

This paper uses individual trial participant data from two large phase 3 dengue vaccine randomised clinical trials to fit a Bayesian survival model of symptomatic dengue illness that is dependent on case-control status (vaccine vs placebo), baseline serostatus, age, and dengue serotype. The main finding is a detailed characterisation of how the relative risk of symptomatic illness changes as a function of time since vaccination and baseline serostatus. As previously reported, children who are seronegative at baseline have a drastically increased risk of symptomatic illness in years 2 and 3 of follow-up. Although this finding is not novel per se, having a precise characterisation of the time-varying risk is important for determining vaccine utility and optimal implementation strategies.

The survival model is fairly complex but this complexity reflects the complexity of the trial data and the inference problem at hand (multiple countries with different proportions of dengue serotypes, active and passive follow-up periods, age-varying effects). I particularly like the use of Gibbs sampling to treat the missing baseline serostatus as a latent variable in the model. This is an elegant method that allows for the analysis of the whole dataset and not restricted to just those with serostatus data (only 10% of participants).

My main comments are about clarity of the manuscript and reproducibility:

1. The Results section needs to explain the big K notation which is fairly complicated (but I can't see any easy way of simplifying it!)

2. The Methods section really needs a section on the data (this could be quite short). Otherwise it's hard to know exactly what sort of data are being used (eg right censoring after first illness is mentioned in the model bit)

3. The Methods section really needs a parameter table. Table 1 gives a summary of model outputs which is useful, but these are not parameters. What I would find useful is a table with all the main model parameters using the same notation as in the text (eg K_0,0_) and the associated prior distributions. This would emphasize that the model has a lot of parameters. This would also be useful to emphasize which parameters were fixed in the model (eg M_b_, sigma?).

4. It says all priors are uniform but this seems a bit odd. Maybe I am misunderstanding something, but the relative risk parameters can take values in (0, infinity) surely? A uniform prior is therefore improper. This is probably not the best choice (uniform priors are often not a good choice).

5. "Model code is available from the authors" and "The model was coded in C++" doesn't really meet *eLife* standards for reproducibility or inspire huge confidence. The paper is entirely based on the validity of the model fitting and this is a complex model that has been hand coded. I fully trust the competence of the authors, but I really think the code should be made available online (via a git repo for example) with a minimum working example for the fitting. If the authors can't make the full dataset openly accessible then a simulated dataset to test the model would be a good alternative.

6. Could you explain a bit more why the parameter M_b_ is fixed at 0.7? It says: "equal to 1 for seronegatives and 0.7 for seropositives, chosen to reflect seropositive participants' reduced disease risk due to their immunity to at least one serotype". I couldn't square this with fact that seropositives also can have increase risk as the secondary infection is more likely to be severe than the first? I get that you don't know how many infections a seropositive individual has had, but I can't really understand the justification for the 0.7.

---

## [Author Response]

Essential revisions:These revisions either reflect a need to temper some of the claims made in the paper or to improve clarity of the work.1) The authors assume in their models that immunity wanes in seronegatives (over 1 year) and is stable in seropositives (maintained over time). It is unclear why these assumptions were made and clarification is needed.

We have now amended our analysis to infer these durations, rather than assume them. The original rationale for fixing these values was that they were imprecisely inferred, although the overall picture was that of short-lived seronegative immunity and long-lived seropositive immunity. Further, it is arguable that three years of follow-up data is insufficient to infer duration. On reflection, we agree with the reviewers that it is more transparent to fit these parameters rather than fix them, and explain where they are imprecisely inferred. We have amended the manuscript throughout to reflect these changes.

2) The authors should adjust the statement "vaccinating seropositives gives greater long-lasting immunity than two natural infections" to reflect the fact that immunity in seropositives may also wane over time in a way that cannot be accurately measured with the current data.

This is a fair point and we have amended the text in the abstract to read, “Our modelling indicates that vaccine-induced immunity is long-lived in seropositive recipients, and therefore that vaccinating seropositives gives higher protection than two natural infections.” to make it clearer. We have also included this point in the discussion text (e.g. lines 614-621) where we are less constrained by word limits.

As our model fits the durations of what we now term “transient immunity”, this conclusion is less dependent on our model assumptions and more dependent on fitted parameter values. We think this conclusion is now sufficiently justified and caveated, and that vaccine benefit to multitypic immune subjects remains an important consideration for CYD-TDV, but please let us know if further amendments are required.

3) On lines 329-330 the authors state "Therefore it may be more beneficial to vaccinate multitypic seropositives than simpler models have predicted [11]." How do the authors know it is the multitypics that benefit as opposed to those with primary immunity before vaccination?

We had not phrased this clearly in the original draft. Under our model, vaccination has two mechanisms: (i) altering disease risk associated with natural infection (denoted by *K_i,D_* values) and (ii) conferring transient immunity (denoted by the *I_bd_* values). For multitypic seropositives, the first mechanism has no effect (as we consider the risk of disease from tertiary or quaternary infection to be equal, i.e. *K_2,D_ = K_3,D_* for either disease type *D*). However, the second mechanism is still predicted to have an effect, and furthermore our results indicate that seropositive transient immunity is long-lived (85% of posterior samples are above 5 years).

Additionally, the model could reject either of these mechanisms depending on the fitted parameter values. For example, seropositive transient immunity (*I_1d_)* values of zero (or alternatively a very short duration) would indicate that only monotypic seropositive vaccinees would benefit, whereas we find positive values for every serotype. Further, we previously ran a sensitivity check where seropositive transient immunity was fixed at zero to see if the data could be explained more parsimoniously, but it could not. However, we were not able to test a model where transient immunity applied only to monotypically immune patients. We have now discussed these points in the results (lines 435-447).

Nevertheless, our results indicate that monotypics will benefit more than multitypics, and benefit to multitypics is dependent on seropositive immunity being long-lived, and we have now included this caveat throughout the text in the abstract (lines 18-19), results (lines 414-417 and 427-447) and discussion (lines 614-621).

4) It would be helpful if the authors further discussed age-effects and their relationship with being multitypic immune.

Please see our response to Reviewer 1’s point (3) below.

5) Please address reviewer 1's concern that the results of this manuscript do not support the claim on lines 311 – 315. Please also address concerns about lines 316-320.

We agree with reviewer 1 and have removed the paragraph from the manuscript.

6) Could the authors comment on whether the fact that serotype effects are more similar than might be expected based on previous articles on these data might relate to the way serotype effects are modeled, or whether this constitutes a major difference from what has been shown previously?

Please see our response to Reviewer 1’s sixth comment below.

7) Please consider reviewer 1's suggestions to improve readability by defining KiD in the main text and revising table 1.

We have revised the notation and text accordingly. We have also adopted the reviewer’s suggestion of a parameter table/model glossary to aid readability. Please let us know if further revisions are required.

8) Line 349 – Where does the value of 0.7 for seropositives come from?

We have now fitted this parameter, and while comparable to what we assumed, it does differ slightly at 0.77 (0.43 – 0.99), and so we are pleased that this issue was raised. Other parameters and conclusions are largely unaffected.

9) Please explain the big K notation in the Results section.

Done.

10) Please add a parameter table to the methods section and add a section on the data.

We have added a parameter table/Model glossary which is now Table S1/Supplementary File 1. We have added a short statement on the data at the start of the Methods. Please let us know if anything further is required.

11) Please justify choice of priors and/or consider sensitivity of results to choice of priors.

Reviewer 2 has commented on our use of uniform priors, specifically regarding relative risks. While in theory these could be unbounded (and therefore improper), in practice this is not a concern (for instance they could be reasonably limited to be less than 100). Further, because we choose as our baseline symptomatic disease among subjects with a single previous infection, all other relative risks (e.g. hospitalized disease among seronegatives) can safely be assumed to be less than 1. We previously used wider relative risk ranges (*Unif*(0,100)), and while resulting parameter estimates were unaffected, it did lead to slower convergence.

We have amended our analysis to allow seronegative transient immunity to be negative, which it now is against serotypes 1 and 2 for 2-5-year-olds, improving the fits to Kaplan-Meier curves this age group and also better demonstrating the risk to seronegatives.

12) Please provide access to the model code. The statement "Model code is available from the authors" does not meet eLife requirements: "Regardless of whether authors use original data or are reusing data available from public repositories, they must provide program code, scripts for statistical packages, and other documentation sufficient to allow an informed researcher to precisely reproduce all published results."

All model code is now available at https://github.com/dlaydon/DengVaxSurvival, together with simulated data that the model can be fitted to. Simulated data was sampled using the parameter posteriors, and preserves comparable case numbers across trial arms, trial phases, age groups, countries, serotypes, disease severities and serostatus. We have also provided example parameter files and pre-compiled binary executables to make running and fitting the model easier.

Reviewer #1:Laydon et al. have conducted an elegant analysis that provides a clear and comprehensive guide to the mechanism of difference of CYD-TDV for dengue virus seropositive vs. seronegative vaccine recipients. The paper includes relevant hypothesis and model schemes as well as figures that show differences between seropositive and seronegative vaccine recipients across a range of covariates. The authors demonstrate, in a clearer way than has been shown before, that CYD-TDV increases disease across all ages in both trials in seronegatives. The authors also show that the enhancing effect is stronger against hospitalized disease than febrile disease. Many of these results have already been presented previously in other papers analyzing the same data, but the modeling approach does provide a new perspective that adds to the story of the CYD-TDV vaccine.Overall, the paper is clearly written and easy to follow, especially given the complexity of the model and subject matter. However, numerous claims are made in the abstract and the discussion that inaccurately reflect the results presented in this paper. In particular, major conclusions of the paper relate to the benefit of the vaccine for seropositive individuals (lines 18-19, 327-328) and an increasing effect of vaccination for seropositive individuals with age (lines 286-290, 329-330). There are potential issues with the modeling approach and how it relates to these conclusions. Further, a discussion about the long-term effects of the vaccine for seronegatives is speculative and problematic (311-320).Comments for the authors:1) Lines 18-19. "Vaccinating seropositives gives greater long-lasting immunity than two natural infections." The claim that fixed-time immunity induced by vaccination of seropositives is maintained at a high level over time is not well supported. The authors assume in their models that immunity wanes in seronegatives (over 1 year) and is stable in seropositives (maintained over time). First, it is unclear why these assumptions were made. The reference for these model choices does not seem to cover immune kinetics over this period of time (Clapham et al. 2016 PLOS Comp Bio). Second, the authors actually tested different assumptions about waning immunity in supplement (lines 667 – 694), which are not described in the main text. In the supplement, the authors demonstrate waning immunity out to 4.5 years (with wide confidence intervals) in both seronegative and seropositive vaccinated individuals. In both cases, the duration of waning is outside of the observational period of the study, and the authors write: "We conclude that three years of follow up data from each patient is insufficient to infer durations." Thus, the authors should adjust the statement "vaccinating seropositives gives greater long-lasting immunity than two natural infections" to reflect the fact that immunity in seropositives may also wane over time in a way that cannot be accurately measured with the current data.

We agree with the reviewer, and so have amended our analysis to fit these durations as opposed to fixing them. While we cannot precisely infer durations, this uncertainty is stated clearly in the results, and the overall picture is that of short-lived seronegative transient immunity and long-lived seropositive transient immunity (lines 427-434).

We also agree with the reviewer’s point that benefit to seropositives beyond two previous natural infections is valid to the extent that seropositive transient immunity is long-lived, and there is some evidence that it is. We have amended the text accordingly in several places to make this caveat explicit (lines 18-19, 414-417, 427-434, and 614-621).

2) Lines 329-330. "Therefore it may be more beneficial to vaccinate multitypic seropositives than simpler models have predicted [11]." How do the authors know it is the multitypics that benefit as opposed to those with primary immunity before vaccination? It would be helpful if the authors could further support this statement by discussing the results that relate to this point. This is especially important in relation to the discussion about dengue rapid diagnostic tests (e.g. lines 23-24), which have very low sensitivity for detecting monotypic DENV immunes.

While one mechanism of vaccine action (change in relative risk) would not affect multitypics, the other mechanism (conferral of transient immunity) would benefit multitypics, for as long as this immunity was present.

We have now made this point more prominent in the results (lines 414-417 and 427-447) and elsewhere. Benefit to multitypics is now supported by fitted parameter values, and not merely values that we had previously assumed.

While our fitted parameters do predict greater immunity to seropositive vaccinees than two previous natural infections, and also that multitypics would benefit, it is true that we were not able to test a model where transient immunity applied only to monotypics (now mentioned in lines 443-447 and 585-587). That said, we believe that potential benefit to multitypics, even though they would benefit less than monotypics, is an important consideration for this vaccine, and we would very much like to leave this consideration in our paper if possible. This is particularly true seeing as we agree with the reviewer that it is difficult to identify monotypic immunes – benefit to multitypics would render this less of an issue, as all seropositives would benefit (we mention this in the discussion in lines 614-627).

3) Lines 286 -290. "Interestingly, fixed-time immunity was found to increase with age in seropositives. In general, heterogeneity between countries' Kaplan-Meier curves can be explained by serotype and seroprevalence, although these factors are insufficient to explain differences in vaccine efficacy by age, for which age-specific effects (independent of serostatus) are required." The conclusion about the benefit of the vaccine even to multitypic immunes seems to be derived from the observation that fixed-effect immunity induced by the vaccine in seropositive individuals increases with age. The authors observe large age-specific fixed-time immunity differences by serostatus (Figure 5), although it is not clear there is much difference across age groups in vaccine efficacy (Figure 4). While the authors state that age specific effects are independent of serostatus, the age effects only occur in seropositives and thus are most plausibly related to more prior infections in children of older ages. It would be helpful if the authors further discussed age-effects and their relationship being multitypic immune.

The predicted benefit to multitypics does not arise from the age-dependence in transient immunity (previously termed “fixed-time immunity”). Instead, it comes from the positive values and long durations of seropositive transient immunity – albeit with the significant caveat that we assumed transient immunity would affect monotypic and multitypic seropositives equally (now mentioned in lines 443-447 and 585-587). The “silent-infection” mechanism does not affect multitypics in our model, because we assume that disease risk from tertiary and quaternary infection is equal (i.e. *K_2,D_* = *K_3,D_*).

We have amended our analysis to allow seronegative transient immunity to be negative (which it is for seronegative 2-5-year-olds against serotype 2), which allows for better fitting of the 2-5-year-old Kaplan-Meier curves. There is now a slight age trend in seronegatives, in that values for 2-5-year-olds differ from older age groups.

Nevertheless, the reviewer’s comment is interesting, and we had not thought of this before. It is possible that the increased transient immunity we estimate for older ages for all seropositives also reflects greater transient immunity in multitypics, although the similar age trend in seronegatives would perhaps argue against this explanation.

Please let us know if we have misunderstood the reviewer’s point. In any case, we have included these points in the discussion (lines 555-561).

4) Lines 311 – 315. "In high transmission settings where children would ordinarily receive at least two natural infections, we predict that even seronegatives would eventually benefit from vaccination, as they would experience one high risk infection, and all subsequent infections would have a much lower risk, in contrast to unvaccinated individuals who would ordinarily experience one moderate and one high risk infection [Figure 1]." The results of this manuscript do not support this claim. While the authors may expect this to be true based on the model scheme (referenced as Figure 1), the results, as currently described, do not prove this point. Further, the caveats provided afterward do not justify inclusion of this paragraph in the discussion (Lines 316-320): "Important caveats include first that disease risk would increase in the short term. Second, and more seriously, a small subset of symptomatic infections are fatal, in which case vaccination could shorten life in seronegatives. Finally, while the eventual benefit to seronegatives is predicted by our model and the model fits well, this remains a prediction yet to be validated by empirical data."). This paragraph should be removed or significantly reworked because it is largely a question of medical ethics.

This is a fair point. We agree have removed the paragraph from the manuscript.

5) Figure 4. It is not clear why vaccine efficacy increases over time in the trial for vaccinated seropositives, even across all age groups. It does not seem to be explained by age effects, waning immunity, or serotype distributions. Is it related to differences in disease outcomes measured in the trial over time (e.g. Figure S8)?

The reviewer is correct: the improvement in efficacy (the decline in hazard ratios) over time for seropositives reflect the better performance of the vaccine against hospitalised or severe disease (at least in as far as this can be inferred given relatively few passive phase case counts), as per what was Figure S8 (now Figure 4—figure supplement 2). We have updated the text accordingly to make this explicit (lines 369-371), and we thank the reviewer for drawing this to our attention.

6) It is surprising that the serotype effects are more similar than might be expected based on previous articles on these data, especially previous observations of greater efficacy against DENV4. Here the authors observe strong enhancement of DENV4 at later timepoints in the trial. Could the authors comment on whether this might relate to the way serotype effects are modeled, or whether this constitutes a major difference from what has been shown previously?

There are some quite large differences in the transient immunity estimates by serotype for each serostatus. For example, in 2-5-year-old seronegatives, transient immunity is -11% (-72% – 46%) for serotype 2, but this rises to 54% (11%-85%) for serotype 4. In 12-16-year-old seropositives, transient immunity is 54% (22%-78%) for serotype 1 and 85% (56%-99%) for serotype 4.

Hazard ratios, particular at later times post first dose, are more similar between serotypes, and this may be due to the fact that we do not model serotype-specific relative risks (and therefore serotype-specific change in relative risk induced by “silent-infection” vaccination). We did attempt to resolve this previously, but the trial data was less well fitted, and there is insufficient power to resolve the parameters, particularly for the passive phase/hospitalised disease. Even here though, the risk enhancement in 2-5-year-old vaccinees is much larger for serotypes 1 and 2 than for serotypes 3 and 4 (either when considering seronegatives only or when considering both seronegatives and seropositives combined).

However, we agree with the reviewer that the greater efficacy previously observed for DENV4 are not as well captured by our analysis. If we had been able to model serotype specific durations of transient immunity, we think that this would have resulted in longer durations for seronegative transient immunity for DENV4, which would likely diminish the DENV4 risk enhancement our main model predicts. Unfortunately however, transient immunity durations that are not serotype-specific are already imprecisely inferred. We have added these limitations to the discussion of the main text in lines 582-598.

7) Table 1 and its explanation in the text are confusing. Instead of a traditional presentation of relative risk, which is described in the text, the table presents the probability of disease expected relative to secondary infections. One option would be to present relative risk estimates in table, and clearly indicate the reference groups in the table. Additionally, while KiD is clearly defined in the methods, it would be helpful to add one sentence to the main text to define KiD, given that the main text uses this notation in multiple paragraphs without explaining its interpretation.

We have made the definition of *K,_iD_* clear in the main text now, and have added in the baseline/reference groups to Table 1 for relative risks and age-specific multiplier of the hazard. That said, we are not sure we understand the reviewer as the relative risks presented in Table 1 are those given in the text. Please let us know if we have misunderstood or if further changes are required.

Reviewer #2:This paper uses individual trial participant data from two large phase 3 dengue vaccine randomised clinical trials to fit a Bayesian survival model of symptomatic dengue illness that is dependent on case-control status (vaccine vs placebo), baseline serostatus, age, and dengue serotype. The main finding is a detailed characterisation of how the relative risk of symptomatic illness changes as a function of time since vaccination and baseline serostatus. As previously reported, children who are seronegative at baseline have a drastically increased risk of symptomatic illness in years 2 and 3 of follow-up. Although this finding is not novel per se, having a precise characterisation of the time-varying risk is important for determining vaccine utility and optimal implementation strategies.The survival model is fairly complex but this complexity reflects the complexity of the trial data and the inference problem at hand (multiple countries with different proportions of dengue serotypes, active and passive follow-up periods, age-varying effects). I particularly like the use of Gibbs sampling to treat the missing baseline serostatus as a latent variable in the model. This is an elegant method that allows for the analysis of the whole dataset and not restricted to just those with serostatus data (only 10% of participants).My main comments are about clarity of the manuscript and reproducibility:1. The Results section needs to explain the big K notation which is fairly complicated (but I can't see any easy way of simplifying it!)

We agree. We have now split up the *K* parameters, which previously denoted too many related but distinct quantities and concepts. As before, *K_i,D_* refers to the relative risk of disease given *i* prior infections of case type *D* (by default active phase/passive phase, or alternatively hospitalised/non-hospitalised or severe/non-severe if considering disease severity). *φ_cD_(α)* refers to the seropositive disease risk where number of previous infections is unknown, and so is a weighted average of *K_1,D_* and *K_2,D_*, for subjects of age *α* in country *c*. Finally, *R_abcD_(α)* is the relative risk of disease of type *D* in stratum *abc*, which encompasses baseline serostatus, aggregate seropositive risk by age *φ_cD_(α)*, and the change in relative risk induced by vaccination (i.e. the vaccine-as-silent-infection model).

These definitions are given in the text and in the model glossary / parameter table. We think the manuscript is clearer as a result, but please let us know if additional changes are required.

2. The Methods section really needs a section on the data (this could be quite short). Otherwise it's hard to know exactly what sort of data are being used (eg right censoring after first illness is mentioned in the model bit)

We have added a short data section at the start of the Methods.

3. The Methods section really needs a parameter table. Table 1 gives a summary of model outputs which is useful, but these are not parameters. What I would find useful is a table with all the main model parameters using the same notation as in the text (eg K_0,0_) and the associated prior distributions. This would emphasize that the model has a lot of parameters. This would also be useful to emphasize which parameters were fixed in the model (eg M_b_, sigma?).

This is an excellent suggestion and we have now added a model glossary (Table S1/Supplementary File 1). This contains fitted and fixed parameters, their prior distributions, as well as various quantities derived from the parameters that should make the methods more readable.

4. It says all priors are uniform but this seems a bit odd. Maybe I am misunderstanding something, but the relative risk parameters can take values in (0, infinity) surely? A uniform prior is therefore improper. This is probably not the best choice (uniform priors are often not a good choice).

While we agree that in theory relative risks could be very large (though not unbounded), in practice this does not align with dengue epidemiology. Since we choose risk of disease among subjects with a single prior infection as our baseline (i.e. *K_1,1_:=1*), relative risks can be expected to be lower than 1. We previously we did run fits where relative risks could vary between 0 and 100. This did not change results but did lead to considerably slower convergence. Further a prior of *Unif*(0,100) would mean that our prior belief considered disease risk in primary or tertiary/quaternary infection to be 99 times more likely to be greater than secondary infection, and this is obviously not the case.

In any case, the priors are not unbounded and are therefore integrable, and so they are not improper. Further, while the hard boundaries of uniform priors can lead to non-symmetrical proposals, we are careful in preserving proposal symmetry. Proposed parameter values that lie outside parameter ranges are “reflected” in the opposite boundary (unfortunately the only analogy I can think of is Pac-Man!).

Please let us know if we have misunderstood your point or if any changes are still required.

5. "Model code is available from the authors" and "The model was coded in C++" doesn't really meet eLife standards for reproducibility or inspire huge confidence. The paper is entirely based on the validity of the model fitting and this is a complex model that has been hand coded. I fully trust the competence of the authors, but I really think the code should be made available online (via a git repo for example) with a minimum working example for the fitting. If the authors can't make the full dataset openly accessible then a simulated dataset to test the model would be a good alternative.

We agree. All model code, together with pre-compiled binaries and example parameter/batch files, is available at https://github.com/dlaydon/DengVaxSurvival. Unfortunately, the underlying trial data is at an individual level and owned by Sanofi Pasteur, and so we are not free to share it. However, as suggested we have provided simulated data, sampled from our parameter posteriors, that has comparable case numbers across serotypes, disease severities and various strata. When the model is fitted on run on this simulated data, it reproduces our results well, albeit with a little noise. We hope that this will be acceptable.

6. Could you explain a bit more why the parameter M_b_ is fixed at 0.7? It says: "equal to 1 for seronegatives and 0.7 for seropositives, chosen to reflect seropositive participants' reduced disease risk due to their immunity to at least one serotype". I couldn't square this with fact that seropositives also can have increase risk as the secondary infection is more likely to be severe than the first? I get that you don't know how many infections a seropositive individual has had, but I can't really understand the justification for the 0.7.

This language was careless on our part, and we thank the reviewer for drawing it to our attention. A seronegative individual, while less likely to become symptomatic upon infection, is susceptible to more serotypes than a seropositive individual, because seropositives have acquired natural immunity to at least one serotype. This susceptibility to fewer serotypes is what the *M_b_* parameter refers to, and we have therefore amended the description of this parameter on lines 130-132.

Nevertheless, we agree with the reviewer, and so we have updated all model runs so that the *M_b_* parameter is fitted rather than fixed. Results are similar (exact values have been updated throughout the text), and the *M_b_* parameter has a mean of 0.77 (0.43-0.99), slightly higher than we had previously assumed. We are grateful to the reviewer for raising this point and we think our manuscript is more robust as a result.